

# Return period of high-dimensional compound events. Part II: Analysis of spatially-variable precipitation.

Diego Urrea Méndez[1], Manuel Del Jesus[1], and Dina Vanessa Gomez Rave[1]

[1]IHCantabria - Instituto de Hidráulica Ambiental de la Universidad de Cantabria, Santander, Spain.

**Correspondence:** Manuel Del Jesus (manuel.deljesus@unican.es)

**Abstract.** This study introduces a comprehensive framework for modeling compound precipitation events, with a focus on handling zero intermittency in rainfall data. By expanding the existing methodologies to a five-dimensional approach and applying the joint return period (JRP) concept using both Gaussian copulas and R-vines with Gaussian, extreme value, and t-Student copulas, we offer a more accurate understanding of these complex events. A key contribution of this study is the
proposal of a model that calculates the multivariate return period in five dimensions, surpassing the commonly used bivariate approach, and considers the dependence of precipitation events across multiple sites, accounting for both lower and upper tail dependencies. The comparison of dependency structures in the generated samples shows that the R-vine structure with extreme value copulas in a multivariate mixed model is particularly effective at capturing the spatial dependence in the data. Our findings emphasize that an inappropriate choice of copulas can lead to either overestimation or underestimation of design
events with defined return periods, with significant implications for risk management.

## 1    Introduction

Understanding natural hazards, such as floods, is crucial for adequately managing the risks associated with these extreme events. These phenomena emerge as a result of complex physical interactions between climatic variables, both across spatial and temporal scales (Zscheischler et al., 2018). A fundamental concept when discussing flood events is that of "compound
events", defined as the combination of multiple drivers and/or hazards that contribute to social and/or environmental risk (IPCC, 2023). These events are responsible for many natural disasters, including floods, making their comprehension, quantification, and analysis of paramount importance.

Compound events manifest in various forms within hydrological systems. A clear example of compound events can be the interaction of precipitation in different areas of a watershed or during diverse temporal periods, leading to floods of
greater magnitude than those predicted by univariate models. Understanding the complex interactions, the underlying physical processes, and the spatial and temporal variability of precipitation is essential for determining the dynamics of hydrological phenomena and their impact on floods. Methodological frameworks for analyzing Compound Events are relatively new; however, many of their components are based on statistical and mathematical methods that have been developed over decades. A recent framework is the one proposed by (Zscheischler et al., 2020), who developed a methodological framework for
typifying and modeling compound events.





Depending on the hydrological scale used, compound precipitation events measured at different rain gauges may contain a considerable number of zeros (zero-inflated data). For example, when considering compound rain events based on monthly maxima, it is common that selecting the maximum event from one station, associated with the maximum of another, includes zero values in the sample. Intermittency in precipitation poses a challenge for capturing dependency structures and correlation

coefficients (Yoo and Ha, 2007).

In this context, various studies (Yoo and Ha, 2007; Ha and Yoo, 2007) have examined the impact of null values on the spatial correlation function of rainfall. To do so, they have segmented the pluviometric records (located in multiple sites) into groups of data with and without zeros, alternating between rain gauge series. These investigations have applied mixed distribution functions (Shimizu, 1993), demonstrating that groups with positive precipitation records obtain useful correlation estimates for

the characterization of rainfall fields(Kedem et al., 1990; Ha and Yoo, 2007).

One of the first mixed models applied in a bivariate approach was developed by (Shimizu, 1993). This approach represents a copula-based mixed distribution function composed of a continuous part (observations greater than zero) and a discrete part (observations at zero). This model has been adapted by various authors (Serinaldi, 2008), overcoming the need to assume an underlying distribution. Several case studies have been developed applying this methodology to bivariate cases (Serinaldi,

2009; Li et al., 2013). Other studies that have considered multiple variables have only modeled intermittency in two variables (Gómez et al., 2018). However, we have not found studies that model zero intermittency in more than two groups or variables, limiting our ability to obtain accurate representations and reliable predictions of complex climate and hydrological processes.

Another fundamental definition when discussing floods corresponds to the notion of the return period (RP). Salvadori et al. (2011) defines the RP as the time elapsed between two successive occurrences of a prescribed event. For example, in the

context of an extreme flood, the RP would indicate how many years, on average, we would expect the flood level to be equaled or exceeded. The concept of RP can be extended to a multivariate context through the Joint Return Period (JRP) and the notion of copula, due to its ability to define multivariate cumulative distribution functions (CDFs).

This concept of JRP has been extensively studied in the literature (Favre et al., 2004; Salvadori, 2004; De Michele et al., 2007; Salvadori and De Michele, 2010; Salvadori et al., 2011; Gräler, 2014; Gräler et al., 2016). However, much of its

development has focused on two-dimensional cases (Zscheischler et al., 2020), with few studies exploring three-dimensional spaces (Grimaldi and Serinaldi, 2006; Pinya et al., 2009; Muthuvel and Mahesha, 2021), given the mathematical complexity and the amount of data required. Recently, Manuel del Jesus et al. (2023) presented a methodological framework that allows the application of JRP calculation to $d$-dimensional spaces. This framework enables the determination of JRP for compound events, such as precipitation measured at multiple sites, but it can also be applied to study the joint behavior of multiple

variables.

In this context, this study has two main objectives: (I) to expand the methodological framework for modeling data with zero intermittency from a bivariate (Shimizu, 1993; Serinaldi, 2008; Villarini et al., 2008), to a five-dimensional approach, focusing on spatially and temporally compounding events (Zscheischler et al., 2020); and (II) to calculate compound events associated with a 100-year JRP, using the methodology proposed by Manuel del Jesus et al. (2023). To achieve this, we employed five

different approaches: Gaussian copulas with and without zero intermittency, and R-vines structures for Gaussian, extreme





value and t-Student copulas. These methods will be applied to model the dependence between five rainfall series. The resulting design events will be compared with those obtained through the traditional univariate method to evaluate the precision and adequacy of each approach.

## 2 Methodology

In this study, we adopt a systematic approach to achieve our research objectives, which include expanding the modeling framework for intermittent zero values and calculating compound precipitations events related to a 100-year JRP. Based on the methodology proposed by Manuel del Jesus et al. (2023), we have modified certain steps to better align with our specific case study. Our methodology consists of the following steps:

1. Diagnosis: This step includes identify the typology, drivers, and scale of compound events.

2. Quantification of compound events: In this step, the methodological framework proposed by Manuel del Jesus et al. is extended to identify compound rainfall events that exhibit spatial and temporal dependence. In these compound rainfall events, it is crucial to consider intermittency (presence of zero values), which is commonly encountered. This consideration leads to the incorporation of multivariate mixed models, which will be detailed in this chapter.

3. Multivariate dependence structures: In this step, we propose and implement various multivariate models. We evaluate these structures based on their ability to capture complex dependencies, considering aspects such as tail dependence, symmetry, goodness of fit, etc. The proposed models will be describe in details in this chapter.

4. Hazard scenario: In this step, Kendall's approach is selected to model dependencies and calculate joint return periods in multidimensional spaces, within the hazard scenario, which can be 'and', 'or', or based on Kendall's method.

5. Determination of the multidimensional return period: We apply the models proposed in step 3 to calculate JRP that account for the spatial and temporal dependence in our data. Specifically, we compute the critical surface for a 100-year JRP event using each of the proposed approaches.

6. Compound design events: In this step, we evaluate the effectiveness of each approach. For this, we compare compound events with a 100-year JRP, using the best-fit model to identify differences in overestimation or underestimation.

While our methodological framework follows the structure proposed by Manuel del Jesus et al. (2023), we have significantly expanded steps 2 and 3 to adapt the method to our specific case study. This expansion allows us to address the challenges of identifying and modeling spatially and temporally compound rainfall events across multiple locations. Furthermore, our comprehensive comparison of different modeling approaches will provide valuable insights into the relative strengths and limitations of each method in capturing the complex spatial and temporal dependencies of rainfall patterns in our study area. In the following sections, we will explain in detail the additions and modifications made to steps.



## 2.1 Exploratory dependency analysis and Intermittency

The methodology for identifying compound rainfall events that exhibit spatial and temporal dependence extends the framework proposed by Manuel del Jesus et al. (2023) through a five-step procedure. Initially, monthly maximum precipitation events are identified for each rain gauge, applying an independence test with a 7-day window (Urrea Méndez and del Jesus, 2023). Subsequently, synchronous compound events are extracted from these identified maxima. To ensure regional representativeness, "regional events" are then selected based on simultaneous rainfall contribution across all stations and, for non-independent events, the highest total precipitation. These events are further validated using flow series, employing a threshold exceeding a 10-year RP and allowing for a 3-day delay to capture the basin's hydrological response. Finally, rainfall events generating flow events with a RP greater than 10 years are retained, ensuring the hydrological significance of the selected compound events. This comprehensive approach aims to identify temporally independent, spatially coherent, and hydrologically relevant compound rainfall events for subsequent analysis within the multivariate mixture modeling framework.

### 2.1.1 Multidimensional mixture model

In the process we proposed for identifying compound rainfall events, it is common to introduce zeros in the selection. When working with these measurements, zeros can obscure clear dependence structures and affect correlation coefficients (Serinaldi, 2009; Li et al., 2013). Given that we are dealing with events that include zeros, the modeling is approached using a mixed continuous-discrete distribution. In this context, we have extended the model proposed by Serinaldi (2008) to describe the joint CDF of a 5-dimensional multivariate mixed model Eq. (1):

$$\Phi(x_1, \ldots, x_n) = P_0 + \sum_{i=1}^{n} p_i F_{X_i}(x_i) + \sum_{i<j} p_{ij} C(F_{X_i}(x_i), F_{X_j}(x_j)) + \sum_{i<j<k} p_{ijk} C(F_{X_i}(x_i), F_{X_j}(x_j), F_{X_k}(x_k)) + \ldots$$
$$+ p_{1\ldots n} C(F_{X_1}(x_1), F_{X_2}(x_2), \ldots, F_{X_n}(x_n)) \tag{1}$$

Where:

- $P_0$ represents the probability mass of zero rainfall at all locations.
- $p_i F_{X_i}(x_i)$ accounts for the probability of rain at only one location $i$, where $F_{X_i}(x_i)$ is the CDF for $X_i$.
- $p_{ij} C(F_{X_i}(x_i), F_{X_j}(x_j))$ describes the joint distribution for pairs of locations, with $C$ being the copula function capturing their dependency.
- $p_{ijk} C(F_{X_i}(x_i), F_{X_j}(x_j), F_{X_k}(x_k))$ represents the joint distribution for three locations.
- $p_{1\ldots n} C(F_{X_1}(x_1), F_{X_2}(x_2), \ldots, F_{X_n}(x_n))$ describes the joint distribution for all $n$ locations, where $p_{1\ldots n}$ is the probability mass associated with rainfall occurring at all locations simultaneously.

## 2.2 Multivariate dependence structure

Considering the structure of the compound precipitation events in our study, we propose the use of multivariate mixed models to address the complexities of the data. As described in equation Eq. (1), we propose modeling these events by groups,





categorizing the data into $2^5 = 32$ distinct groups based on the presence of non-zero values across the five series (see Fig. 1). These groups range from scenarios with only one station recording rainfall (such as in group 1, among others) to all five stations registering precipitation simultaneously (group 32). In this chapter, we present five modeling approaches designed to capture the spatial and temporal dependencies within these groups, while effectively handling the frequent occurrence of zero values characteristic of the data.

1. Gaussian Copula without intermittency (Gaussian): This approach considers the joint dependence between compound rainfall events without accounting for zero intermittency, including all rainfall data without exception.

    2. Gaussian Copula with zero intermittency (Gaussian groups): This approach considers the dependence between rainfall compound events using non-zero groups, modeling the dependence between 2 to 5 series. This method models each group using the Gaussian copula, leveraging its ability to model high dimensions.

3. Vine Gaussian copulas (Vine Gaussian): This approach utilizes R-vine structures with Gaussian copulas to model all pairs of series. It combines the flexibility of R-vines for capturing complex dependence structures with the efficiency of Gaussian copulas for pairwise modeling.

    4. Vine extreme copulas (Vine extreme): This approach uses R-vine structures with a diverse set of bivariate copulas, including Student t, Clayton, Gumbel, Joe6, BB1, BB6, BB7, BB8, and independent copulas. Rotated versions of
135        Archimedean copulas are included to model negative dependence.

    5. Vine T-student copulas (Vine t-student): This approach uses R-vine structures with t-student copulas to model all pairs of series. It combines the flexibility of R-vines for capturing complex dependence structures with the efficiency of t-student copulas for pairwise modeling.

The selection of these approaches was based on the copulas' ability to model compound events in high dimensions, tail
dependencies, and symmetries. Additionally, the univariate model was included to conduct a comparative study, evaluating the inherent advantages and disadvantages of applying each approach. Figure 1 presents a schematic that visually describes the models.

The selection of the Gaussian copula was not based on its ability to model extremes, given the limitations identified by Jaser and Min (2021). Instead, it was chosen for its capacity to model high dimensions and with the objective of contrasting the
results derived from its implementation in modeling extreme events. This choice is supported by its frequent application in climate and hydrological research focused on simulating extreme conditions (Chen and Guo, 2019).

## 2.3 Study site and information sources

Our research has been conducted in a basin located in northern Spain (basin area: 465 km²), following the course of the Besaya River. This geographical choice was based on the influence of rainfall on the characteristic flooding patterns of this region. To
carry out our analysis comprehensively, we have selected 5 strategically distributed rain gauge stations, belonging to the State




**Figure 1.** Schematic representation of five approaches for modeling compound rainfall events using multivariate mixture models. The diagram illustrates how each method (Gaussian, Gaussian groups, Vine Gaussian, Vine extreme and Vine t-Student) applies multivariate mixture model concepts to five-dimensional compound rainfall data across locations (G1-G5). It depicts the grouping strategy for rain/no rain patterns and the progression from univariate to 5D copula models, highlighting differences in modeling spatial dependencies and zero intermittency within the multivariate mixture framework.

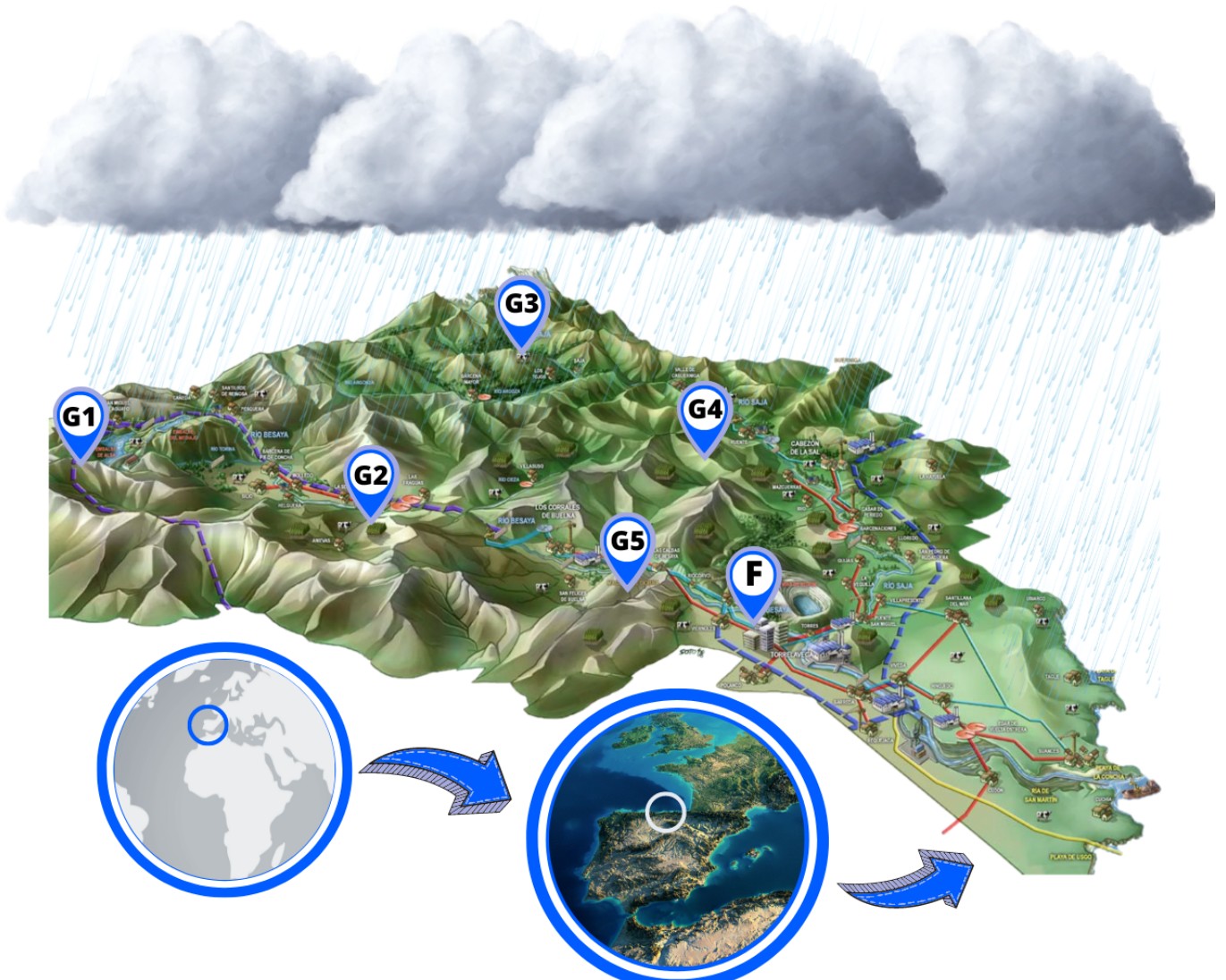

3D Map, Government of Cantabria <http://www.municipiossostenibles.com/>

**Figure 2.** Location of the project area - Besaya River Basin, Cantabria, Spain. The rain gauge stations, represented by the letter G (G1, G2, G3, G4, G5), are marked in blue, indicating their geographic location within the basin. The streamflow gauge station is represented by the letter F, also in blue. The figure provides a detailed 3D map of the area, highlighting key hydrological monitoring points essential for the study.

Meteorological Agency (Aemet) of Spain. Additionally, we have used flow series from a station located in the lower part of the basin. Figure 2 shows the location of the project area, where the selected rain gauge stations are presented in blue and labeled with G, while F represents the selected flow station.





The study utilizes daily rainfall and flow data from 1950 to 2019, providing a comprehensive view of long-term climatic
and hydrological patterns. A rigorous quality control process was implemented, including outlier identification (Gonzalez and
Bech, 2017), review of repeated values, and application of quality indices to handle missing data, null values, and false zeros
(Llabrés-Brustenga et al., 2019; Lez-Rouco, 2001). This process ensured the reliability and consistency of the dataset for
subsequent analyses.

## 3  Results

### 3.1  Diagnostics


Our analysis identified the typology of the compound events in this study as spatially and temporally compounding storm
events, aligning with the classification proposed by Zscheischler et al. (2020). The drivers were determined to be heavy
rainfall events, classified as climatic drivers, generated by storms and measured at five rain gauges within the catchment. The
scale definition encompassed both temporal and spatial dimensions, considering various time periods and multiple locations
within the catchment. This approach revealed that precipitation patterns across different connected locations and times led to
aggregated impacts. The resulting hazard was identified as flooding, with flood damage as the final impact. Figure 3 provides
a schematic representation of the compound event analysis conducted in this study, illustrating these findings.

### 3.2  Exploratory dependency analysis and Intermittency

Based on the definition of typology, drivers, and scale, we conducted the selection of compound events, considering their
spatial and temporal dependence. This process allowed us to identify clear correlation structures between events by groups
of non-zero values. All selected events were associated with flow events that exceeded a 10-year RP threshold, representing
potential flooding events in the study basin. Figure 4 illustrates the selection process applied to our case study.

Considering the intermittent presence of precipitation (zero values) and the implications this entails (Yoo and Ha, 2007;
Ha and Yoo, 2007), we employed multivariate mixed models. Based on this structure, the precipitation records were divided
into 32 groups, on which the subsequent stages of the study were developed. Given the considerable number of groups and
to simplify the interpretation of the findings, we will focus on the group where rainfall occurs simultaneously in all stations
(group 32 - Fig. 1). In parallel, a general analysis of the results obtained from the other groups will be provided.

### 3.2.1  Pre-treatment of data

The data preprocessing was carried out in two stages. In the first stage, we focused on identifying tied values, as precipitation
data often presents challenges due to the occurrence of dry days or zero values, which can result in potential ties (Tootoonchi
et al., 2022). Dividing the data into groups proved to be an effective solution for overcoming this limitation. In the second
stage, we checked for the presence of additional duplicate values, such as those produced by drizzles, which may have resulted
in tied records.





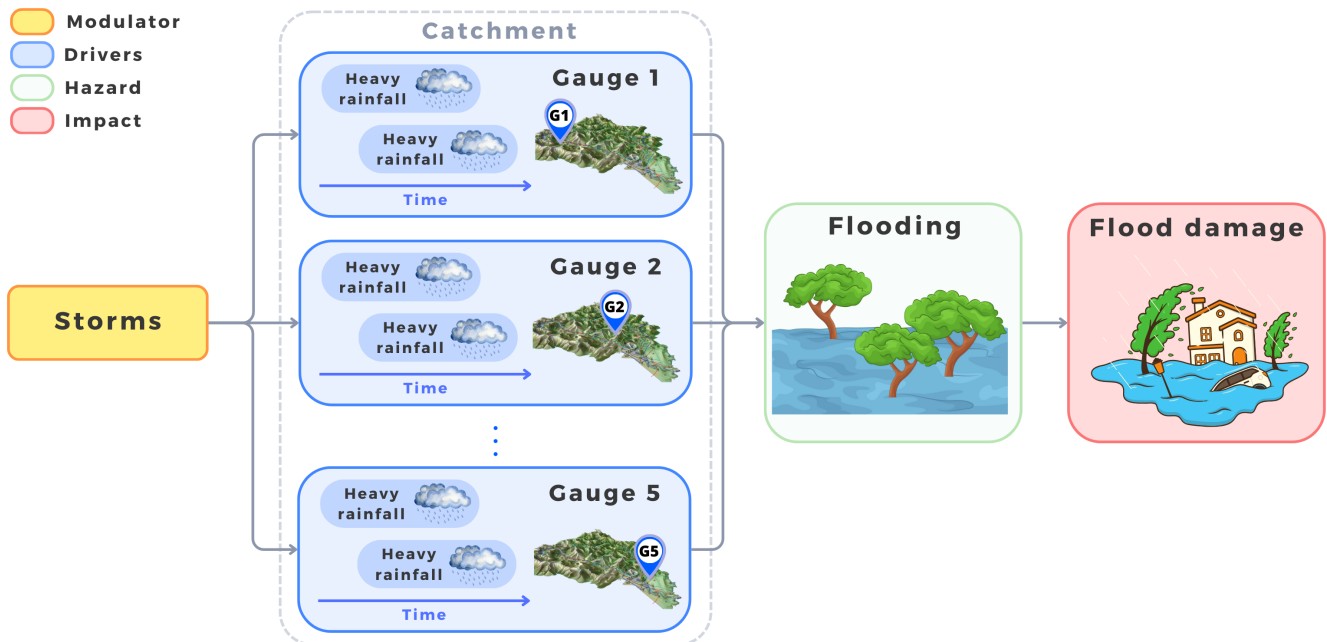

**Figure 3.** Schematic representation of a spatially and temporally compounding flood event. Storms (modulator) generate heavy rainfall (drivers) at multiple locations within a catchment, measured by five rain gauges over time. The accumulation of rainfall leads to flooding (hazard), ultimately resulting in flood damage (impact). This diagram illustrates the spatial and temporal dependencies considered in the study for calculating joint return periods of extreme precipitation events.

In the second stage, we verified whether the selected events exhibited autocorrelation. To analyze this, we applied the autocorrelation function (ACF) to the series selected by season and by group. Figure 5 presents the autocorrelation plots calculated for group 32. When analyzing the autocorrelation plot of the five event series, it is observed that there is no significant correlation between values at different time intervals. This suggests that the data do not show autocorrelated patterns, implying that past observations do not have a direct influence on future observations.

When exploring the autocorrelation functions in the remaining groups, similar results were observed. These functions provided a consistent perspective across all cases, highlighting that the studied series lacked any indication suggesting the presence of autocorrelated patterns. This finding reinforces the notion that precipitation observations in these groups are distributed independently over time, which is crucial for the future stages of analysis.

### 3.2.2 Analysis of variable dependencies and Zeros

To evaluate the dependency of the selected events, Kendall's correlation coefficient was used for all pairs of stations. In Figure 6, the results obtained for group 32 are presented. In the upper triangular matrix, Kendall's $\tau$ values are displayed in a heatmap, where positive dependencies between events across all stations can be observed. In the lower triangular matrix, scatter plots





**Figure 4.** Methodology for the selection of compound precipitation events. In step 1, we select independent monthly maximum events. In step 2, synchronous events associated with the events identified in step 1 are extracted. In step 3, independent events are selected, considering regional events. In step 4, we validate the selected events using flow series with a threshold exceeding a 10-year return period. In step 5, we finally select the compound events.





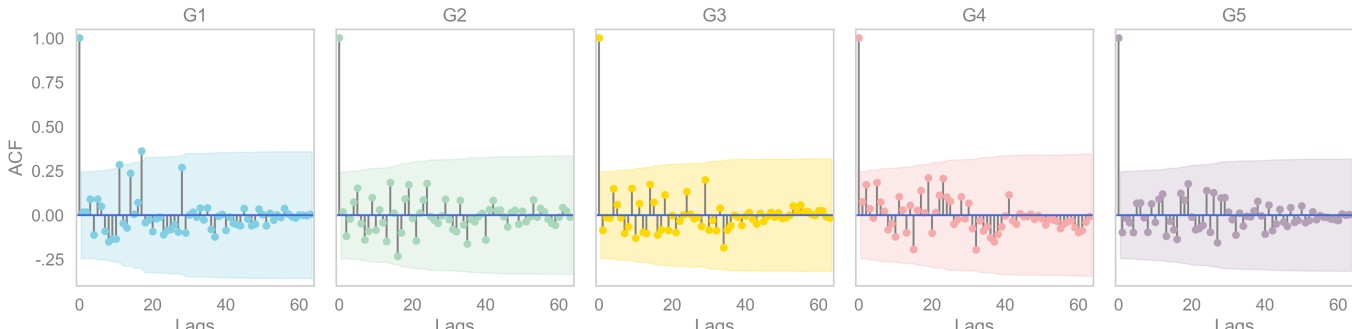

**Figure 5.** Autocorrelation diagrams for precipitation series with simultaneous rainfall across all stations associated with group 32.

show the joint behavior of the variables in pairs. It is interesting to note that the distribution of points in most plots shows a higher density in the upper tail and, in a few cases, in the lower tail, suggesting the existence of possible tail dependence. Additionally, it is important to mention that the data presented in these plots were previously transformed from the original

variable space to the standard uniform space.

As in other studies (Brunner et al., 2018), the estimator by (Schmidt and Stadtmuller, 2006) was applied to determine tail dependencies, with the limitations associated with this method (Serinaldi et al., 2015). The results from this indicator showed that both upper and lower tail dependence were present in the data. However, it is also worth considering that upper tail dependence might be expected, as previous research has demonstrated that extreme precipitation events exhibit upper tail

dependence (Serinaldi, 2008; Evin et al., 2018).

The removal of dry days resulted in generally stronger correlations among the selected compound events. When exploring the correlation functions in each of the analyzed groups, various behaviors were observed, with positive correlation coefficients found in most groups and both upper and lower tail dependencies present in the data. In some specific cases (trivariate groups), tail dependence was not as evident, and slightly weaker correlation coefficients were obtained (0.2-0.4).

### 3.2.3   Marginal distributions

The adjustment of extreme values is generally approached using two different methods: (I) employing the generalized extreme value (GEV) distribution in a block maxima context, and (II) using a generalized Pareto distribution (GPD) within the peaks-over-threshold framework (Coles, 2001). Since the method for selecting sets of synchronous rainfall events did not fit a specific approach, it was necessary to choose the methodology that best suited the data. The results revealed that the sets of events

at individual stations and for each group were adequately fitted to the GEV distribution. One selection and fitting criterion corresponded to the AIC, while the other involved assessing the fit in the upper tail of the distribution.

The parameters of the marginal GEV distributions were determined using maximum likelihood estimation and Bayesian fitting techniques. Comparisons were made between the results obtained from both approaches, showing similarities in the scale and shape parameters. To ensure a fair comparison, the mean value of the distributions associated with each parameter,





**Figure 6.** In the lower triangular matrix, scatter plots are presented, showing the joint behavior of the events for pairs of stations in group 32. The upper triangular matrix displays the corresponding Kendall correlation coefficients, best represented with a heatmap. Positive association patterns are visible in all cases.





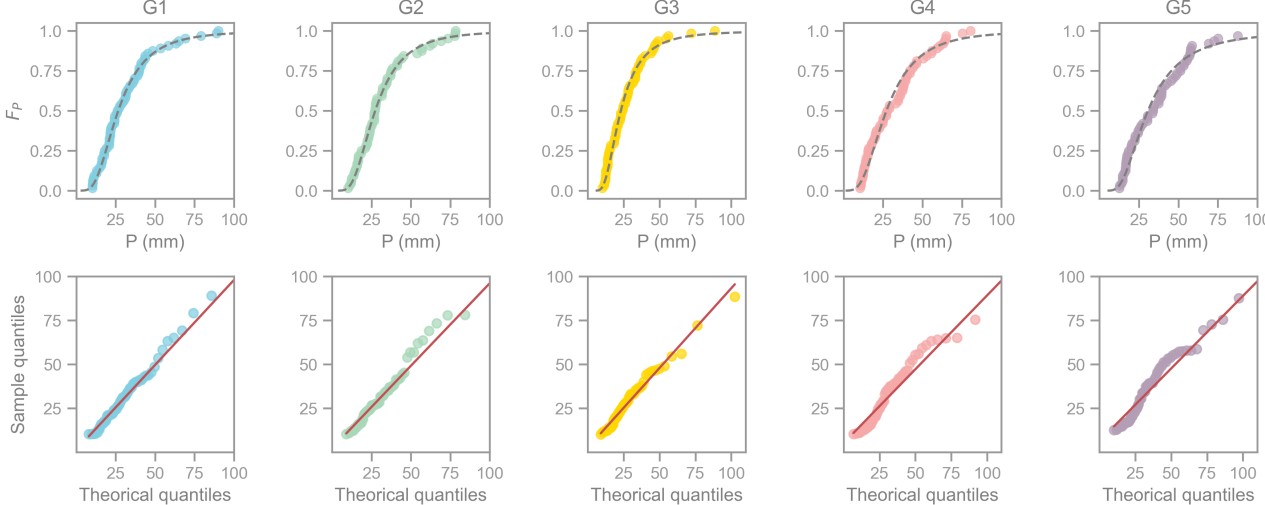

**Figure 7.** The top row shows the cumulative distribution functions (CDFs) for rainfall compound events selected for G1 to G5 in group 32, with the dashed line representing the theoretical distribution. The bottom row displays the quantile-quantile (QQ) plots, comparing observed quantiles to theoretical ones; the red line indicates perfect correspondence. Deviations from the red line highlight differences between the observed and theoretical distributions.

obtained through Bayesian analysis, was used. Although slight differences were observed in the shape parameter, the results indicated that the Bayesian analysis more accurately fit the data. Using the mean values of the parameters associated with each distribution, CDFs were constructed for the stations selected for rainfall compound events in group 32. The results can be seen in the first row of Fig. 7. The results are shown in the first row of Fig. 7, while the second row presents the quantile-quantile (QQ) plots, generated to evaluate the behavior of the tails.

Following the described procedure, the data were fitted to the GEV distribution in the other groups. When fitting the distribution functions to the GEV using the data from the other analyzed groups, reasonable fits were obtained. Additionally, the QQ plots for each group were checked, and it was observed that the GEV adequately represented the tail behavior.

### 3.3 Multivariate dependence structures

The dependency structures analyzed include the approaches described in the methodological chapter of this study. The selection
criterion applied to compare the performance of the proposed approaches corresponds to goodness-of-fit tests and the comparison between the observed and modeled dependency structures. The results obtained in this chapter will not limit further analyses; they will only serve as a basis for comparing the performance of the proposed approaches.





**Table 1.** Comparison of methods based on logLik, AIC, and ECP values

| N. | Enfoque | logLik | AIC | ECP/CvM | ECP/Pvalue | ECP2/CvM | ECP2/Pvalue |
|----|---------|--------|-----|---------|------------|----------|-------------|
| 1 | Gaussian | -12753.93 | 25527.86 | | | | |
| 2 | Gaussian groups | -350.80 | 721.60 | | | | |
| 3 | Vine Gaussian | 94.60 | -169.20 | 1.53 | 0.59 | 0.069 | 0.941 |
| 4 | Vine extreme | 105.60 | -185.30 | 1.38 | 0.65 | 0.045 | 0.996 |
| 5 | Vine t student | 101.40 | -162.80 | 1.43 | 0.62 | 0.048 | 0.994 |

### 3.3.1 Copula parameter estimation and goodness of fit

To compare the proposed approaches, the parameters were estimated using the pseudo-maximum likelihood method. A semiparametric
method was used because it is based on the rank transformation of the variables and does not require assumptions about
the marginal distributions. The approaches where R-vine structures were used were estimated using the automated model
estimation and selection technique from Dißmann et al. (2013).

For the evaluation of the proposed approaches' performance, graphical methods and different statistical goodness-of-fit tests
were used. The tests applied were the Likelihood function, Akaike Information Criterion (AIC), and the Cramér-von Mises
test. For scenarios including R-vine copulas, the goodness-of-fit test based on the empirical copula process (ECP) was used,
with the Cramér-von Mises test statistic and a bootstrap with 1,000 replications to calculate the p-value. In the Gaussian copula
scenario, the p-value was also estimated using the Cramér-von Mises test and bootstrap. The results for group 32 are presented
in Table 1.

Table 1 presents the goodness-of-fit results for the different approaches under study, including the Cramér-von Mises test
(ECP/CvM), the associated p-values (ECP/Pvalue), and the Akaike Information Criterion (AIC). These metrics evaluate the
quality of the model fit, with the ECP/CvM indicating the discrepancy between the empirical and fitted copulas, and the AIC
providing a balance between model fit and complexity. Higher p-values indicate better model adequacy.

The Vine extreme approach performs best, with an Akaike Information Criterion (AIC) of -185.30 and a log-likelihood
(logLik) of 105.60, indicating a superior balance between fit and complexity compared to the other models. Additionally, it
provides a better fit to the dependency structure of the data, as shown by its Empirical Copula Process - Cramér-von Mises
(ECP/CvM) value of 1.38, which is lower than the other approaches evaluated.

The p-values from the Empirical Copula Process (ECP/Pvalue) test indicate that all approaches, including Vine extreme,
were valid at a 0.05 significance level, reinforcing the appropriateness of the models tested. In summary, Vine extreme not only
optimizes the fit in the first level of dependency but also in the second level, with a Second-Level Empirical Copula Process -
Cramér-von Mises (ECP2/CvM) value of 0.045, positioning it as the most efficient model for the data analyzed.

To analyze the results for the remaining groups, a box plot was constructed, as presented in Fig. 8, where the distributions
of AIC for all groups in each proposed approach are compared. Note that approach 5 was not included because its fit value is





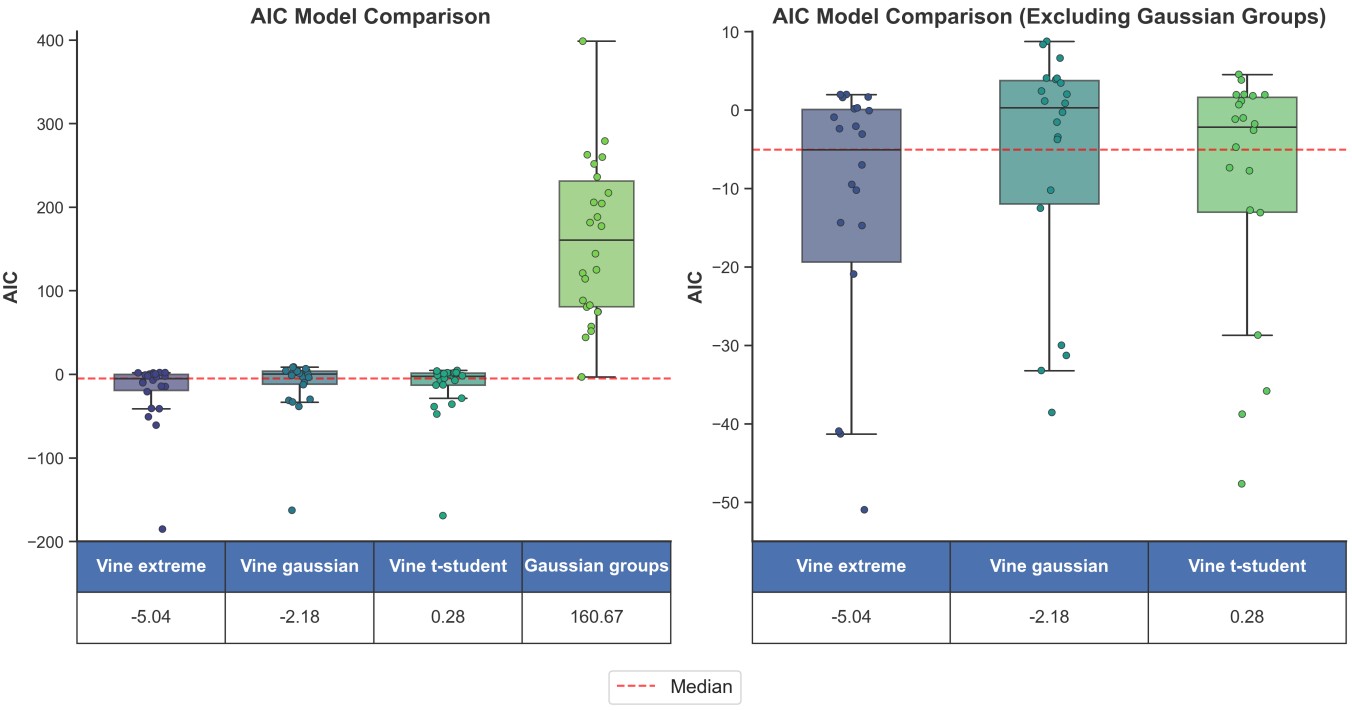

**Figure 8.** Boxplots represent the results in terms of model fit and number of parameters using the AIC criterion to compare the proposed approaches. On the left, the results of approaches 1 to 4 are shown. On the right, results of approaches 1 to 3 are displayed, excluding approach 4, to better compare the models with lower AIC values. Note that approach 5 is not included, as its fit is based on a single point and it performs very poorly, distorting the scale of the plot.

a single point and it performed poorly, which would distort the scale of the graph. Additionally, the results for group 32 have been omitted, as they were analyzed previously (Table 1), allowing for a clearer comparison of the Vine approaches.

In the left box plot of Fig. 8, the results of the proposed scenarios are presented, showing that the Gaussian groups scenario had the lowest performance compared to the R-vine scenarios. The right box plot provides an expanded view of the left plot, excluding the Gaussian groups scenario.

To facilitate this comparison, the lowest median AIC value is indicated by a red dashed line. The median was selected to account for data variability and the influence of outliers in the box plots. The results indicate that the Vine extreme approach

achieved the lowest median AIC value compared to the other approaches. It is important to note that the results for approach 1 (Gaussian) were not included in Fig. 8 because it does not involve mixed multivariate models (groups) and is thus represented by a single value. The AIC obtained by approach 1 was 25527.86, making it the highest value and, therefore, the worst-performing approach.





### 3.3.2 Evaluating the dependence structures

Based on the goodness-of-fit tests, the efficiency of the best-performing (4) and worst-performing (1) approaches was compared using graphical methods with the results from group 32. Two key factors were considered: The first was crucial for assessing whether the dependency of the observed values was maintained, reduced, or improved. The second evaluated how well the tail dependencies of the observed data were reproduced.

The results of the first factor are shown in Figure 8. In the lower triangular matrix, scatter plots illustrate the joint behavior of the observed variables. In the upper triangular matrix, kernel density estimate (KDE) plots, generated from 1,000 samples 275 using the R-Vine extreme approach, are displayed. Each subplot includes Kendall's $\tau$ (black rectangle), with red rectangles highlighting the differences between Kendall's $\tau$ for the observed and simulated data. In most cases, the simulated data achieved a higher Kendall's $\tau$ than the observed data, with a maximum difference of 0.06. This indicates that the R-Vine extreme copulas effectively captured and reproduced the correlation structures of the observed series.

To assess the second factor, a graphical evaluation of tail dependencies was performed using KDE plots. In this case, the 280 density of the simulated series is represented by color intensity, with darker and denser tones indicating a higher concentration of data in that region. In Figure 9, it can be observed that the simulated data show greater density in the upper tails, and in some cases, in the lower tails as well. This finding supports the ability of the copulas used to accurately capture and reproduce the behavior of the real variables in terms of their extremes and dependencies.

Based on the results from approach 1, Fig. 10 was developed, following a structure similar to the one previously described. 285 However, the plots in the upper triangular matrix reflect the performance of this approach. In this case, the maximum differences between correlation coefficients reach 0.17, indicating inferior performance compared to approach 4 in terms of reproducing correlation structures. Additionally, it is clear that the generated data lack density in the upper tail, implying that the regions with higher concentration do not exhibit a pronounced symmetry and are distributed across various areas, including the lower 290 tail density.

Based on the analyzed results, the Vine extreme approach demonstrated its ability to reproduce upper tail dependencies. In the other analyzed groups, similar results were obtained, where both lower and upper tail dependencies were observed in the simulated samples.

### 3.4 Hazard scenarios

295 The mathematical definitions for the JRP and critical layer in $d$-dimensional spaces were developed using the methodology described by Manuel del Jesus et al. (2023). In this study, we focused on identifying multivariate design events with a 100-year RP, associated with the $Kendall$ hazard scenario for the proposed approaches.

### 3.4.1 Kendall function

Following the steps outlined in (Manuel del Jesus et al. 2023) and based on the methodology developed by (Salvadori et al., 300 2011), we first simulated a sample large enough to obtain reliable estimates of the critical probability level $t$ of Kendall. While





**Figure 9.** In the lower triangular matrix, scatter plots are presented, showing the joint behavior of the observed variables. In the upper triangular matrix, kernel density estimate (KDE) plots are shown, created from 1000 samples of the best-fitting approach in terms of AIC and tail dependence (R-Vine extreme). All subplots include Kendall's $\tau$ (black rectangle) calculated for both the observed and simulated data. The upper matrix also includes red rectangles highlighting the differences between the Kendall's $\tau$ of the observed and simulated data.





**Figure 10.** In the lower triangular matrix, scatter plots are presented, showing the joint behavior of the observed variables. In the upper triangular matrix, kernel density estimate (KDE) plots are shown, created from 1000 samples of the worst-fitting approach in terms of AIC and tail dependence (Gaussian without intermittency). All subplots include Kendall's $au$ (black rectangle) calculated for both the observed and simulated data. The upper matrix also includes red rectangles highlighting the differences between the Kendall's $au$ of the observed and simulated data.





(Serinaldi, 2013) suggests simulating at least 10,000 samples for the bivariate case, our results indicated that larger samples were necessary as dimensionality increased. For the most critical case of 5 dimensions, we found that 1 million data points yielded reliable results.

For each set of values, we calculated the corresponding copula value. In the case of vine copulas, where a closed mathematical definition of the joint distribution function is not available, we applied the Monte Carlo method to integrate the joint probability function.

Given that calculating the copula value for all simulated samples is computationally expensive, especially in the case of vine copulas, we adopted the methodology of (Manuel del Jesus et al. 2023) based on supervised learning called Gaussian Processes (GPR) (Rasmussen and Williams, 2006). By implementing this method, the time required to calculate the joint distribution function was simplified. For this, 10,000 samples were calculated using the Monte Carlo method. With these samples, the GPR model was trained, thus allowing the subsequent evaluation of the distribution function in the remaining set of samples.

For the training phase, 80% of the data was used, while 10% was assigned to the testing phase and another 10% was reserved for model validation. Regarding the kernel that performed best in the models, it was identified that the Rational Quadratic kernel was the most suitable. The predictions generated by the GPRs demonstrated a remarkable ability to capture the underlying relationships in the simulated data. In terms of accuracy, both the relative error and the absolute error, calculated during the validation phase, remained at levels close to 0.01 in the best-performing model and 0.05 in the worst-performing model, across all analyzed groups.

From the obtained results, we calculated the Kendall function and obtained the critical probability level $t$ for each scenario. Due to the low computational cost of GPRs, it was feasible to apply bootstrap methods and compare them with the results associated with large samples. The results showed that choosing a sufficiently large sample size ($10^6$) for the construction of the Kendall function is equivalent to taking the average value of the Kendall functions generated through the bootstrap method. A broader discussion of the topic can be found in (Serinaldi, 2013). Figure 11 presents the results obtained from the Kendall curves for each approach studied in group 32.

To calculate the critical level $t$, it was necessary to calculate the 100-year RP. Considering that we have more values per year than in the case of annual maximum, the quantiles in this case move to the extreme part of the distribution. Note also that each Kendall function is calculated from the continuous part of the function described in Eq. (1), that is, it considers the complete CDF.

Considering the goodness-of-fit results and the tail dependencies of the approaches, it can be analyzed that the copulas with upper tail dependence had a better fit and a higher critical probability level $t$. The best-performing approach (4) obtained a critical value of 0.993, while the lowest-performing approach (1) obtained a critical value of 0.778.







**Figure 11.** Kendall curves for the 5 approaches analyzed, applied to group 32. Additionally, the value of $t$ associated with the quantile corresponding to a 100-year return period is presented.

## 3.5 Multidimensional return period

### 3.5.1 Critical layer

Having defined the critical probability level $t$ associated with a 100-year JRP and the joint distribution function for each approach, we proceeded to generate the corresponding critical layer. It is important to note that this process was carried out

independently for each of the proposed approaches.

For each approach, we generated 1 million synthetic events. These synthetic events represent possible real occurrences in our multidimensional space. This approach, while computationally intensive, was made both viable and efficient through the use of pre-trained Gaussian Process Regression (GPR) models. The implementation of GPR allowed us to effectively handle the large volume of necessary calculations, leveraging the advantages of this advanced machine learning method.





For each set of synthetic events, we calculated the joint distribution function using the GPR models. We then selected those event combinations that corresponded exactly to the critical level $t$, thus defining the critical layer for each approach. This procedure was iterated until we obtained a sufficient number of events on the critical layer for each approach. Iteration was necessary because, given the specific nature of the critical level $t$, only a small fraction of the synthetic events would correspond exactly to this value.

### 3.5.2   Most Probable Design Event


In this chapter, we present the results of applying two methods for selecting design events in a multivariate context (Manuel del Jesus et al. 2023) (Salvadori et al., 2011). The first method was based on computational optimization techniques to identify a single most probable design event, while the second focused on selecting a set of design events.

In the first method, we employed the Spotpy library (Houska et al., 2015) in Python to select maximum probability events.
The objective was to maximize the probability density function, and to achieve this, various optimization algorithms were evaluated. The Maximum Likelihood Estimation (MLE) algorithm offered the best performance compared to other available algorithms. However, in dimensions greater than 3, the accuracy of the solution obtained by this method showed a significant decrease.

Solving this loss of precision in high dimensions was easy because we had sufficient event combinations on the critical layer.
For each combination of events, we calculated the density function and selected the one with the highest density. Compared to the described optimization procedure, this represented an advantage in terms of precision, although it was computationally more expensive.

For the second method, we applied the Metropolis-Hastings method, a widely recognized technique for generating samples from probability distributions when direct sampling is not possible. In this context, we used the joint density function of the
copula ($f_{XY...W}$) as the target probability distribution. The procedure consisted of repeated iterations: in each iteration, we selected a set of candidate events within the critical layer and then calculated the ratio between the probability density of these candidates and the probability density of the current event. The decision to accept or reject these candidates was based on acceptance probabilities. By repeating this process multiple times, we obtained a sample of potential design events that captured the variability of events within the critical layer.

### 3.6   Application to inland flooding

In this chapter, we compare multivariate design events with design events calculated from univariate analysis. To do this, it was necessary to calculate, using the univariate method, the precipitation associated with a 100-year RP for each rain gauge. For the univariate method, due to the size of the watershed (>1 km²), it was necessary to apply a spatial rainfall reduction factor (Román, 2022; Invias, 2009). The calculated reduction coefficients depend on certain hydro-morphological characteristics of
the watershed, such as the concentration time ($t_c = 14.01h$) and the area. The computed coefficients are presented in Table 2.

To compare the results of univariate and multivariate analysis, it was necessary to calculate the average precipitation in the watershed using both approaches. For univariate approach the average precipitation was computed using Thiessen polygons,





**Table 2.** Spatial reduction methods and corresponding $K_A$ values

| Spatial reduction method | $K_A$ |
|---|---|
| Temez | 0.82 |
| Institution of Civil Engineers, Proceedings | 0.58 |
| Fhrüling | 0.20 |
| North America | 0.34 |

considering the influence area of each rain gauge. This procedure was replicated using the most probable events from the set of multivariate events for each of the proposed approaches. In the multivariate approach, the blue bar represents the method with the best fit in terms of AIC. For the univariate approach, the blue bar indicates the coefficient that best fits the study area, as the Temez coefficient is widely used in Spain (Román, 2022). The results of this analysis are presented in Fig. 12.

Figure 12 presents precipitation estimates for a design event with a 100-year RP. Among the multivariate methods, significant differences are observed, with estimates ranging from 46.49 mm (Gaussian) to 66.1 mm (Vine t-student). The Vine Extreme model, which shows the best fit according to AIC, estimates a mean precipitation of 63.45 mm. Compared to this method, the Gaussian, Gaussian Groups, and Vine Gaussian models tend to underestimate the events, while the Vine t-student overestimates them.

When comparing Vine Extreme with univariate methods, Temez (61.42 mm) comes closest, although it still slightly underestimates the event. This comparison underscores the importance of carefully selecting the spatial rainfall reduction method in univariate approaches

Figure 13 extends this analysis by comparing all possible events, not just the one with the highest probability density. In this broader context, it is evident that all univariate methods, when compared to Vine Extreme, consistently underestimate the mean precipitation. This observation highlights the importance of considering multivariate approaches to adequately capture the variability and spatial and temporal dependence of extreme precipitation events, especially in contexts where these characteristics are significant.

When using the set of multivariate events obtained through the Metropolis-Hastings method, instead of the combination of the most probable events, the results displayed in Fig. 13 are obtained. The Kernel Density Estimate (KDE) plots shown at the top of Fig. 13 were generated using the precipitation values calculated from the multivariate event set for each approach. The blue KDE highlights the approach with the best performance in terms of goodness of fit. As in the previous case, the results show that the design events from approaches that did not fit the observed data well tend to overestimate precipitation values. In the lower section, the univariate design events are presented again for comparison. The results suggest that considering more than one event can provide a better understanding of the problem. In this case, our findings indicate that multivariate events may result in significantly higher magnitudes compared to the univariate analysis, as seen in the case of the 'Vine Extreme' approach, which captures higher precipitation values in the upper tail.



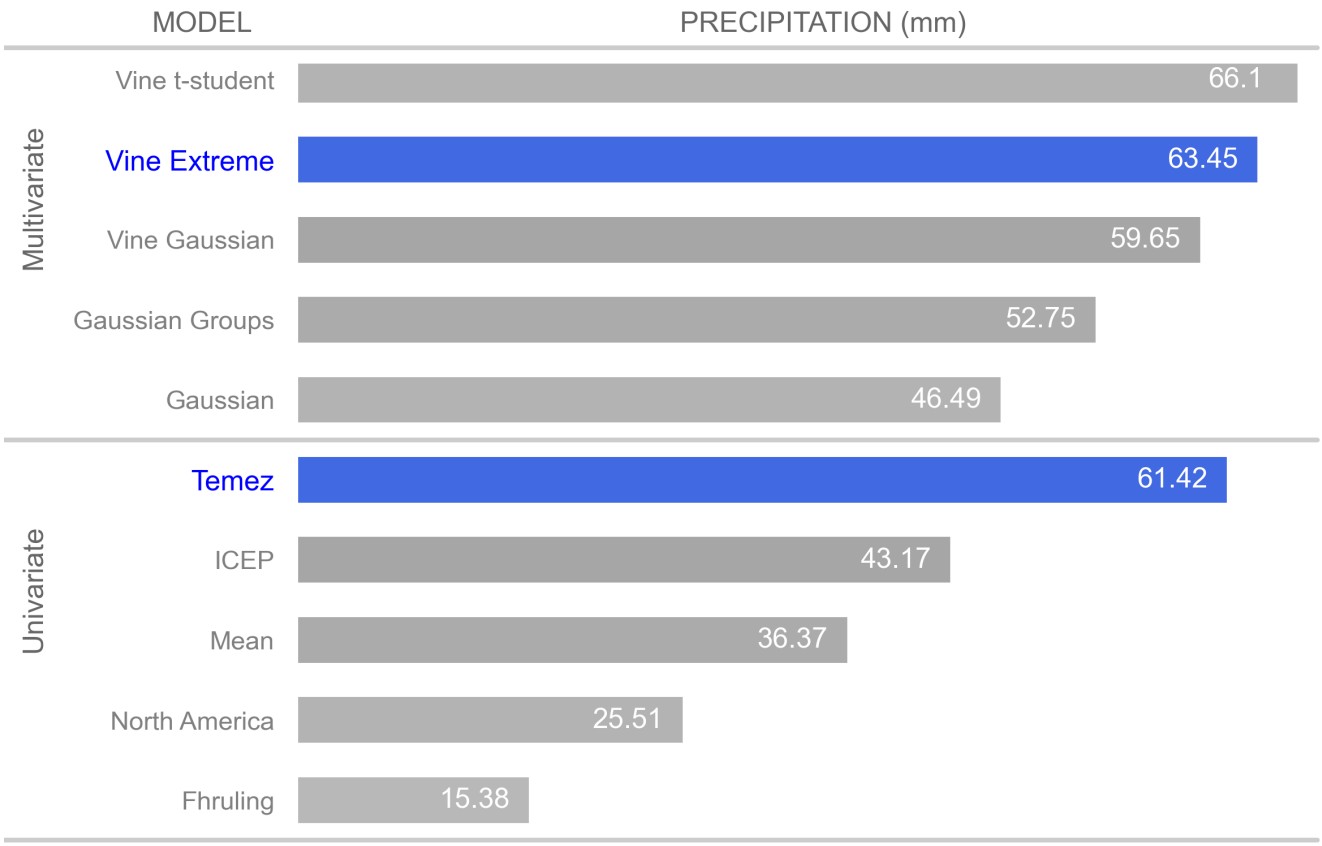

**Figure 12.** In the upper bar chart, a bar graph is shown representing the mean precipitation, calculated from the most probable multivariate design events for each of the proposed scenarios. In this diagram, the blue bar represents the approach that achieved the best fit in terms of accuracy, while the gray bars correspond to approaches with lower performance. In the lower bar chart, a bar graph is shown representing the mean precipitation calculated from the univariate design events, considering various applied rainfall reduction coefficients. The blue bar represents the coefficient that best fits the study area.

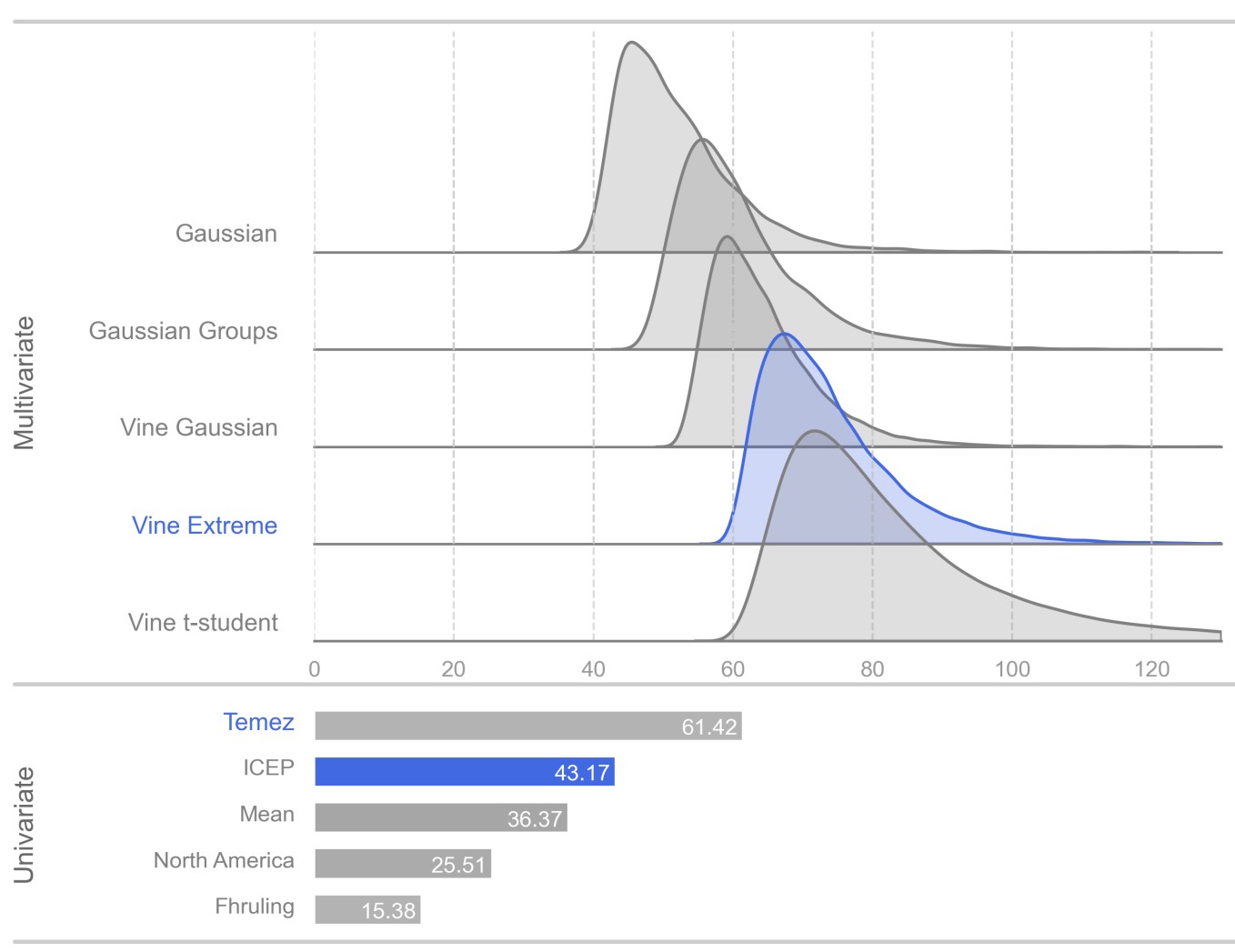

**Figure 13.** In the top figure, the Kernel Density Estimate (KDE) plots are presented, representing the set of multivariate events on the critical layer for each proposed approach. The events correspond to a JRP of 100 years. The blue KDE represents the approach with the best performance in terms of goodness of fit. In the bottom bar chart, the average precipitation calculated from the univariate design events is shown, considering various applied rain reduction coefficients. The blue bar represents the coefficient that best fits the study area.





## 4 Discussion

The first objective of our study was to expand the methodological framework for modeling data with zero intermittency, extending it from a bivariate approach to a case of multiple dimensions (5 dimensions). This transition allowed us to describe the physical process, taking into account the interactions between precipitation stations, in order to understand the underlying patterns and behaviors in our data. By incorporating a greater number of dimensions and addressing zero intermittency, we were able to capture a more comprehensive picture of the phenomena under study.

To achieve the second objective, it was necessary to address the methodology and results related to the Kendall hazard scenario. For this, we applied advanced approaches for each proposed method, supported by previous research (Salvadori et al., 2011), which allowed us to obtain reliable estimates of Kendall's critical probability level $t$. As dimensionality increased, we faced several challenges. Initially, we observed that as dimensionality grew, it became necessary to simulate larger sample sets to ensure the accuracy of the Kendall function. This finding was supported by the implementation of Gaussian Processes

(GPR), which accelerated the calculation of the joint distribution function, demonstrating its effectiveness in improving the precision of our estimates. Additionally, we performed a comparison between the required sample size and the results obtained through the bootstrap method. The results revealed that using very large sample sizes is equivalent to taking the average value of the Kendall functions generated through the bootstrap method. This underscores the importance of increasing the sample size in higher dimensions to ensure the reliability of the results.

Considering the goodness-of-fit results and the tail dependencies of the approaches, we found that the better the fit, the higher the critical probability value $t$. The findings highlighted the importance of the model selection stage in multivariate models, as selecting a model that does not fit the data well can either underestimate or overestimate Kendall's critical probability values. The notion that the Gaussian copula is used for simplicity is flawed and can have serious consequences if it does not fit the data properly.

Regarding the critical layer, the need to increase its resolution as dimensionality increases is emphasized. This is particularly crucial in the most complex case, with 5 dimensions, where simulating a large number of samples, even in the order of millions, is required. However, our results once again showed that with the implementation of Gaussian Processes (GPR), a large number of events can be generated with a reasonable computational cost. The methodology used provided a significantly more accurate description of events with a JRP of 100 years.

Furthermore, this study presented two methods for selecting multivariate design events. One was based on computational optimization techniques, while the other used the Metropolis-Hastings method. However, it was observed that the optimization method showed a significant decrease in accuracy when applied to higher dimensions (above 3), making it less recommended in such cases. The solution to address this loss of accuracy was relatively simple, as long as an adequate set of event combinations on the critical layer was available. In this context, we evaluated the density function for each event combination and selected

the one with the highest density. Compared to the previously described optimization procedure, this approach represented a significant advantage in terms of accuracy.





The results obtained in this study provide valuable insight into the importance of selecting appropriate approaches for hydrological analysis of precipitation-related design events. Our findings indicate that the multivariate approach demonstrates greater accuracy in estimating precipitation compared to the univariate approach, suggesting that considering the joint dependence between variables, specifically precipitation, can significantly improve the quality of the results. Moreover, the underestimation of precipitation magnitude by approaches that did not fit the observed data well highlights the importance of precision in multivariate analysis, especially in extreme weather event scenarios. Our study also underscores the value of using multivariate events, obtained through methods such as Metropolis-Hastings, for a better understanding and risk assessment.

## 5 Conclusions

This study introduced and explored various methodologies with the aim of providing a more comprehensive and accurate understanding of the physical processes related to compound precipitation events. First, we significantly expanded the methodological framework by addressing events that include zeros in precipitation data, representing a fundamental advancement in modeling these phenomena. Additionally, we implemented the methodology developed by [Manuel del Jesus et al. 2023] to characterize the multivariate return period using the Kendall function, comparing and contrasting five different approaches.

Our findings highlight the crucial importance of selecting the appropriate copula approach when modeling compound events, as an inadequate choice can lead to the overestimation or underestimation of design events with defined return periods, which has significant implications for risk management and planning related to precipitation. In this study, the most suitable model for capturing spatial and temporal dependence was the multivariate mixture model based on vine copulas with extreme value copulas to represent pairwise behavior.

Although the compound events selected in this study exhibited both upper and lower tail dependencies, this is not always the case, which underscores the importance of conducting a thorough exploratory data analysis and considering various copula models to select the best fit for the data. The selection of an appropriate method for conducting a hydrological study plays a critical role in the accuracy of precipitation design event estimates, particularly in the context of compound climate events.

The results obtained throughout this research have provided solid and consistent evidence in favor of the multivariate approach, which outperforms the univariate approach in capturing uncertainty and dependence between variables. Comparing the results of the univariate analysis with the best-performing multivariate approach clearly shows that the univariate approach tends to underestimate design events, which can have significant repercussions for decision-making related to water resource management and infrastructure planning in the study region. Finally, the inclusion of multivariate events, obtained through methods such as Metropolis-Hastings, has proven to be a valuable tool for improving our understanding of risks associated with extreme precipitation. These findings have significant implications for decision-making in water resource management and provide a solid foundation for future research in the field of hydrology.



*Author contributions.* Manuel and Diego conceptualized the study and developed the theoretical framework. Diego also implemented the computational models and performed the data analysis. Dina and Diego designed and conducted the statistical tests. All authors contributed to the interpretation of results and the manuscript preparation.

*Competing interests.* The authors declare no competing interests.

*Disclaimer.* No potential conflict of interest was reported by the authors.

*Acknowledgements.* This work received financial support from the Government of Cantabria through the Fénix Programme and from Grant RTI2018-096449-B-I00, funded by MCIN/AEI/10.13039/501100011033 and by "ERDF A way of making Europe." Additionally, the author used artificial intelligence tools to review the English. These tools assisted in optimizing the clarity of the text, without influencing the
scientific content or analysis presented.





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
