# Peer review of "Return period of high-dimensional compound events. Part II: Analysis of spatially-variable precipitation."

_Hydrology and Earth System Sciences, 2024_

## Referee Comment (RC1)

**REVIEW REPORT**

**Journal:** HESS
**Paper:** hess-2024-335
**Title:** Return period of high-dimensional compound events. Part II: Analysis of spatially-variable precipitation
**Author(s):** Manuel Del Jesus, Diego Urrea Méndez, and Dina Vanessa Gomez Rave

**GENERAL COMMENT.**

For the reasons given below, I recommend a **REJECTION (WITH POSSIBLE RESUBMISSION).**

**SPECIFIC COMMENTS.**

**Line(s) 26–27**

**AUTHOR(s).** Depending on the hydrological scale used, compound precipitation events measured at different rain gauges may contain a considerable number of zeros (zero-inflated data).

**REFEREE.** From a Statistical point of view, it is a nightmare, and the problem is far from being resolved. The introduction of a mixed model, and then considering only positive rainfall values, does not seem to fix the question, for introduces other problems (see my comment below).

**Line(s) 36–38**

**AUTHOR(s).** One of the first mixed models applied in a bivariate approach was developed by (Shimizu, 1993). This approach represents a copula-based mixed distribution function composed of a continuous part (observations greater than zero) and a discrete part (observations at zero).

**REFEREE.** The model by Shimizu (1993) is not copula-based: the modeling via copulas is present nowhere in the paper, also considering that it deals with a mixture of discrete-continuous distributions. Incidentally, none of the marginals used in that paper is heavy-tailed, possibly inadequate to deal with rainfall extremes.

**Line(s) 43–44**

**AUTHOR(s).** Another fundamental definition when discussing floods corresponds to the notion of the return period (RP). Salvadori et al. (2011) defines the RP as the time elapsed between two successive occurrences of a prescribed event.

**REFEREE.** The true novelty of the paper by Salvadori et al. (2011), beyond the mathematical formalization of a multivariate notion of RP, and the introduction of original multivariate design techniques, is that the calculation of the RP is written in terms of copulas only in any multi-dimensional setting (not only bivariate).

**Line(s) 56–57**

**AUTHOR(s).** In this context, this study has two main objectives: (I) to expand the methodological framework for modeling data with zero intermittency from a bivariate (Shimizu, 1993; Serinaldi, 2008; Villarini et al., 2008), to a five-dimensional approach...

**REFEREE.** Is 5 a magic number, in hydrology or elsewhere? Increasing the dimension is not a synonymous of novelty: what about if tomorrow I publish a paper on a 6-dimensional approach? It may solve problems in 6 dimensions, but may not change the paradigms... in addition, you only dealt with a sub-class (Group 32) of the 5 dimensional problem...

**Line(s) 72–73**

**AUTHOR(s).** This consideration leads to the incorporation of multivariate mixed models, which will be detailed in this chapter.

**REFEREE.** The mixed model ignores/spoils the correlation structure of the sequences of $(0, > 0)$'s in the rainfall time series, and in turn the COMPOUND nature/feature of the events.

For example, the total precipitation in the two sequences A and B below is the same (0 means no rain, 1 means rain), but the COMPOUND impact of series B could be devastating as compared to the one of series A:

A=[0,1,0,1,0,1,0,1,0,1]

B=[1,1,1,1,1,0,0,0,0,0]

Here, the correct approach would be to use a stochastic renewal process, as in

G. Salvadori and C. De Michele. Statistical characterization of temporal structure of storms. Advances in Water Resources, 29(6):827–842, 2006. doi: 10.1016/j.advwatres.2005.07.013

**Line(s) 77**

**AUTHOR(s).** 4. Hazard scenario

**REFEREE.** A reference to

Salvadori, G., Durante, F., De Michele, C., Bernardi, M., and Petrella, L.: A Multivariate Copula-Based Framework for Dealing with Hazard Scenarios and Failure Probabilities, Water Resources Research, 52, 3701–3721, https://onlinelibrary.wiley.com/doi/abs/10.1002/2015WR017225, 2016

should be put here: it is the first paper where Hazard Scenarios are first formalized in terms of Copulas, including those mentioned by the Authors.

**Line(s) 80**

**AUTHOR(s).** Specifically, we compute the critical surface...

**REFEREE.** Surface or hyper-surface, with dimension larger than 2?

**Line(s) 94–96**

**AUTHOR(s).** To ensure regional representativeness, "regional events" are then selected based on simultaneous rainfall contribution across all stations and, for non-independent events, the highest total precipitation.

**REFEREE.** This sentence is quite obscure: what do you mean by "for non-independent events"? What are the events you are considering? Monthly maxima at different stations? What kind of independence do you consider? Spatial-Pairwise? Spatial-Global? Note that they are different: Pairwise independence may not imply Global one... How do you test it? How do you identify a "homogenous" region? Via clustering procedures?

**Line(s) 105 & Eq. (1)**

**AUTHOR(s).** In this context, we have extended the model proposed by Serinaldi (2008)...

**REFEREE.** I do not think it is an extension, except perhaps for the dimension, but then it would be trivial: you simply try to account for the probability of mutually exclusive events in dimensions larger than 2, nothing too special that could justify a specific paper...

$\Phi$ is not precisely defined, what should it represent? A joint CDF? Eq. (1) looks like a linear combination of probabilities: what is its meaning? What is the domain/support of the parameters $p$'s, and their inter-relationships? No information/explanation is provided...

Furthermore, written in this way, the formula may account twice for the probability of zero rainfall: in fact, by definition, $F(x) = P(X \leq x)$ (in whatever form you write it, for one or more variables) includes the case that the variable(s) take on the value 0, even if the threshold $x$ is strictly larger than 0: the probabilities in Eq. (1) look more like conditional ones. In addition, $P_0$ should depend upon the location, I would be surprised if it were the same at all stations.

In all cases, you should prove that $\Phi$ is a genuine probability distribution, which however suffers from over-parametrization (i.e., the number of $p$'s): estimating all these parameters in a high-dimensional space is a torment, at least from a numerical standpoint, for the estimates almost certainly never correspond to optimal values (at best, they are suboptimal in the most favorable cases).

**Line(s) 125–126**

**AUTHOR(s).** Gaussian Copula without intermittency (Gaussian): This approach considers the joint dependence between compound rainfall events without accounting for zero intermittency, including all rainfall data without exception.

**REFEREE.** The presence of 0's yields Ties, which adversely affect (spoil) statistical techniques: how do you manage such a problem, given the fact that no effective solutions are present in Literature? The failure of the Gaussian approach may be due to the fact that, as remarked below, it is inadequate for dealing with extremes, but also to the fact that Ties play against it in this approach: you must make things clear.

**Line(s) 128–129**

**AUTHOR(s).** This method models each group using the Gaussian copula, leveraging its ability to model high dimensions.

**REFEREE.** Unfortunately, a Gaussian framework is unsuitable for dealing with maxima, such as those considered in this paper. . .

**Line(s) 130–132**

**AUTHOR(s).** This approach utilizes R-vine structures with Gaussian copulas to model all pairs of series. It combines the flexibility of R-vines for capturing complex dependence structures with the efficiency of Gaussian copulas for pairwise modeling.

**REFEREE.** I am not sure that a Gaussian copula could be "efficient" (whatever you mean with this unspecified feature) if the true dependence structure is not Gaussian itself. A Gaussian copula has feasible mathematical properties, but also strong limitations, especially considering Extreme phenomena, as abundantly pointed out in Literature.

**Line(s) 133**

**AUTHOR(s).** Vine extreme copulas (Vine extreme): This approach uses R-vine structures with a diverse set of bivariate copulas. . .

**REFEREE.** Perhaps, dealing with maxima, Extreme Value copulas should better be used, but these exclude the case of negative dependencies: a justification is required here.

**Line(s) 143–146**

**AUTHOR(s).** The selection of the Gaussian copula [. . . ] is supported by its frequent application in climate and hydrological research focused on simulating extreme conditions (Chen and Guo, 2019).

**REFEREE.** This sentence/explanation sounds like a suicide, for it reads as: since (inexperienced) practitioners frequently use the Gaussian copula, this justifies its use, and therefore we use it. No comment.

**Line(s) 149–150**

**AUTHOR(s).** To carry out our analysis comprehensively, we have selected 5 strategically distributed rain gauge stations...

**REFEREE.** What do you mean by "strategic"? Do you mean "representative" of the hydrological regime (whatever the word "representative" could mean)? What regionalization/clustering procedures/criteria did you use to decide that these are "strategic"? Or these stations are the only ones available (and so the number 5 has a justification)?

**Line(s) 155–156**

**AUTHOR(s).** A rigorous quality control process was implemented, including outlier identification (Gonzalez and Bech, 2017), review of repeated values...

**REFEREE.** What do you mean by "review of repeated values"? And rigorous with respect to what benchmarks?

**Line(s) 175–177**

**AUTHOR(s).** Given the considerable number of groups and to simplify the interpretation of the findings, we will focus on the group where rainfall occurs simultaneously in all stations (group 32 - Fig. 1).

**REFEREE.** So what? You introduce a tricky model, then you realize it is too complex, and thus you use the simplest case given by Group 32: essentially, it corresponds to a classical "AND" hazard scenario. The fact that the model is a mathematical mess was already clear in Eq. (1), so why not considering the case of Group 32 directly, which simplifies the discussion, as well as the mathematical treatment. In practice, you boasted about solving a problem in 5 dimensions, but then you only dealt with a specific sub-case.

**Line(s) 185–187**

**AUTHOR(s).** Figure 5 presents the autocorrelation plots calculated for group 32. When analyzing the autocorrelation plot of the five event series, it is observed that there is no significant correlation between values at different time intervals.

**REFEREE.** To the best of my understanding of the plot, quite a few estimates of the ACF are outside a (traditional) 5% Confidence Band, and thus I would suspect that the data ARE auto-correlated...

**Line(s) 195**

**AUTHOR(s).** In the upper triangular matrix, Kendall's $\tau$ values are displayed in a heatmap...

**REFEREE.** Confidence Intervals for the estimates must also be provided.

**Line(s) 203–204**

**AUTHOR(s).** The results from this indicator showed that both upper and lower tail dependence were present in the data.

**REFEREE.** Believe me, with such data you cannot really claim anything about the possible (statistical) presence of Tail Dependence: this is just visual statistics, too often a deceiving practice used by inexperienced practitioners...

**Line(s) 226–227**

**AUTHOR(s).** Additionally, the QQ plots for each group were checked, and it was observed that the GEV adequately represented the tail behavior.

**REFEREE.** Formal Monte Carlo Goodness-of-Fit tests, and the corresponding p-values, would be less visual and more objective (e.g., Kolmogorov-Smirnov, or even better Anderson-Darling ones).

**Table 1**

REFEREE. In Table 1, GoF p-values are missing for the first two cases, they should be shown.

**Line(s) 256–257**

AUTHOR(s). To analyze the results for the remaining groups, a box plot was constructed, as presented in Fig. 8, where the distributions of AIC for all groups in each proposed approach are compared.

REFEREE. You must first check that the model is admissible via a GoF test, and then (and only then) select the "best" model (according to some criterion) ONLY among the admissible ones. The plots of the AIC's alone in Fig. 8 are of little interest/significance: the corresponding models could all be non-admissible without, first, carrying out suitable GoF tests.

**Line(s) 271–272**

AUTHOR(s). The first was crucial for assessing whether the dependency of the observed values was maintained, reduced, or improved.

REFEREE. How you could "improve" a dependence remains a mystery to me...

**Line(s) 283–284**

AUTHOR(s). This finding supports the ability of the copulas used to accurately capture and reproduce the behavior of the real variables in terms of their extremes and dependencies.

REFEREE. Statistically speaking, at most you can hope it: your conclusions are only based on visual analyses, be careful.

**Line(s) 291**

AUTHOR(s). Based on the analyzed results, the Vine extreme approach demonstrated its ability to reproduce upper tail dependencies.

REFEREE. You should add: assuming it is really present.

**Line(s) 324–327**

AUTHOR(s). To calculate the critical level t, it was necessary to calculate the 100-year RP. Considering that we have more values per year than in the case of annual maximum, the quantiles in this case move to the extreme part of the distribution. Note also that each Kendall function is calculated from the continuous part of the function described in Eq. (1), that is, it considers the complete CDF.

REFEREE. This claim is quite obscure, and should be clarified. Intuitively, it should be enough to properly set the constant $\mu$ in the definition of the Kendall RP to fix the right temporal scale (e.g., $\mu = 1/12$). However, this looks like a Kendall RP conditional to the fact that rain is present.

**Line(s) 329–330**

AUTHOR(s). The best-performing approach (4) obtained a critical value of 0.993, while the lowest-performing approach (1) obtained a critical value of 0.778.

REFEREE. Assuming that these results make sense, you should interpret them, and discuss the consequences.

**Line(s) 341–344**

AUTHOR(s). This procedure was iterated until we obtained a sufficient number of events on the critical layer for each approach. Iteration was necessary because, given the specific nature of the critical level t, only a small fraction of the synthetic events would correspond exactly to this value.

**REFEREE.** Frankly speaking, I really doubt that any of the events generated actually lies on the critical layer, if only for the sake of numerical approximation. Most likely, you have fixed some tolerance coefficient: you must clearly explain how you accept that an event lies on the critical layer.

**Line(s) 354–355**

**AUTHOR(s).** Solving this loss of precision in high dimensions was easy because we had sufficient event combinations on the critical layer. For each combination of events, we calculated the density function and selected the one with the highest density.

**REFEREE.** It is not clear what you mean by a "combination of events", and how it is chosen. What is its sample size and how is it decided? What is its density function (the joint one?) More details must be given for the sake of discussion and reproducibility.

**Line(s) 371–372**

**AUTHOR(s).** To compare the results of univariate and multivariate analysis, it was necessary to calculate the average precipitation in the watershed using both approaches.

**REFEREE.** Average precipitation could have little to do with the Extreme Value approach: I understand that it is part of common hydrological practice, but then it seems that the Authors are playing at the same time on two different layers, as if they were trying to have a foot in both camps. A justification is required here.

**Line(s) 379–381**

**AUTHOR(s).** Compared to this method, the Gaussian, Gaussian Groups, and Vine Gaussian models tend to underestimate the events, while the Vine t-student overestimates them.

**REFEREE.** Here, as well as below, you cannot speak about under- or over-estimates: this makes sense only if you know the true value. Here you can only speak about relative smaller/larger values.

**Line(s) 399–ff.**

**REFEREE.** The Discussion and the Conclusions sections could/should be merged in a single section "Discussion & Conclusions".

---

## Community Comment (CC1)

*Larhyss Journal, ISSN 1112-3680, n°60, Dec 2024, pp. 293-296*
© 2024 All rights reserved, Legal Deposit 1266-2002

**DISCUSSION**

**EVALUATING THE EFFECTIVENESS OF THE EXISTING FLOOD RISK PROTECTION MEASURES ALONG WADI DEFFA IN EL-BAYADH CITY, ALGERIA**

**By**

***BEN SAID M., HAFNAOUI M.A., HACHEMI A.,
MADI M., BENMALEK A.***

Published in Larhyss Journal, No 59, September 2024, pp. 7-28

**Discusser**

***KHEBIZI H.***

Water resources mobilization and evaluation laboratory, The National Higher School of Hydraulics, Blida, 09000, Algeria

h.khebizi@ensh.dz

This discussion aims to offer more importance to the original work carried out by Ben Said et al. (2024). The article entitled "Evaluating the Effectiveness of the Existing Flood Risk Protection Measures Along Wadi Deffa in El-Bayadh City, Algeria" focuses on the floods in the wilaya of El Bayadh notably the overflowing from the wadis into nearby urban areas. The prolonged lack of rainfall has increased human activities encroaching upon the dry areas near the wadis. This paper focuses on the October 1, 2011, event in El Bayadh. The event was numerically replicated through hydrological and hydraulic modelling using HEC-HMS and HEC-RAS software. The integration of the influence of key structures such as protective walls, channels, and buildings as determinant factors shows the originality of the modelling process. This enhanced model incorporates real-world complexities, leading to a more accurate representation of the flood scenario. Utilizing the calibrated model, the performance and capacity of the channel and protective structures in safeguarding the city's nearest buildings were assessed. The obtained results demonstrated that the October 1, 2011 occurrence, with a peak discharge of 425 m$^3$/s, greatly exceeded the channel's capacity, which can only handle a peak discharge of 180m$^3$/s.

During the 2000s, a drought was recorded in the Algerian territory due to lack of rainfall, as the result of climate change. In the Saharan areas, torrential floods were marked, including Adrar, Ghardaia, Béchar, Naama and El Bayadh. In these areas, floods appear periodically where the periodicity changes in time and space. In addition, the flood depends on various factors: part of it runs off, part evaporates, and part infiltrates the ground. The amount of infiltrated water depends on the nature of the soil, in particular its

permeability, its slope and its vegetation cover, which can slow down the flow of water and promote its infiltration. However, vegetation can be neglected in large parts of the wadis in the Sahara. In addition, slope variation and rainfall intensity will also play a role in the proportion of infiltrated water to runoff water. The flow in the watercourse over an extended period depends on the volume of water following a significant inflow, particularly heavy rains in the upstream areas of the wadi.

Saharan environments play a role in the storage of water due to the sandy material nature. During exceptional floods, these environments have a hydrological role when quantities of water exceed the usual quantities. The areas on the edge according to the surrounding topography play a role in the animation of the flow and its overflow. The overflow of water in neighbouring urban areas is done in a direction of urban growth to the detriment of the natural environment of the Oued without taking into account the limits not to be crossed, in particular the areas supplying the Oued. The morphology of the watercourse also depends on the sediments that it carries and the geology of the region, affecting in particular the slopes of the banks and the shape of the flood zones. A good understanding of the change in time and space of fluvial forms and sediment flows is very useful as a preventive measure when carrying out urban area expansion projects. For this, the study of aerial photographs allows, for example, to quickly observe geomorphological adjustments of a watercourse over time and thus it can offer more indication about vulnerable zones that can be affected by the sediment flows. Consequentially, analysis of sedimentary flows transiting through the Saharan system, from the source zone of a watercourse, through the transit zone, then finally in the accumulation zone becomes necessary for vulnerability mapping.

In the Saharan zones, the absence of vegetation on the banks helps to maintain the flow speed where the flow speed is much higher in the centre of the wadi bed compared to that of the edges. It is important to mention that the application of hydrology and hydraulics concepts alone does not allow us to fully understand all the physical parameters that govern floods. For this, the integration of the simple practical technique of the trinity of Leeder in the flooding modeling can offer more significance to the deduced scenario. Leeder's technique is a model that aims to represent river dynamics (Taylor, 2014; AGRCQ, 2024). It includes the interrelations between three main components: the structure of the flow, the transport of sediments, and the development of the forms of the bed.

Concerning the flooding evolution in time and space, the sediment transport component can also be described at several scales in the Saharan region. On an annual scale, we can compare the volumes of sediment transported as bed load compared to the volume transported in suspension by a watercourse. We can determine the morphological change caused by the transported sediment. On an event scale, we can look at the volume of sediment transported during a flood and, above all, at the time during the flood when the maximum transport occurs. On a sediment scale, we can look at the role of sediment size in terms of their probability of being carried as bed load or in suspension.

The management of urban extensions depends on the geomorphology of the wadi bed. Management recommendations vary according to the scale of the spatial analysis, in particular the scale of the watershed (profile along the Oued), the scale of a section (main channel), the scale of the bed, and therefore the shape created by the sediments. The increase in speed causes an increase in the size and volume of sediments and has a direct implication on the agglomerations, which are built along the wadi and/or sometimes in the beds of the wadi. The sediments transported create either erosion forms or accumulation forms that cause changes in the geomorphology of the wadi and its course over time and consequentially have a direct impact on current or future extensions. For this, the hydrogeomorphological diagnosis based on hydrosedimentary dynamics, the geomorphological trajectory, and the dynamism of a watercourse over time becomes primordial for a flooding modelling study.

**REFERENCES**

AGRCQ (Association of Regional Watercourse Managers of Quebec) (2024). Guide to the management of Quebec waterways, Watercourse dynamics, Chapter 3, Quebec, Canada, 63p. https://agrcq.ca/guide-gestion-cours-eau   (In French)

TAYLOR O. (2014) A watercourse: it's alive! Geomorphology and fluvial dynamics, Agro-environmental day 2014, Mauricie agro-environmental consultation table, Trois Rivières, Quebec, Canada, November 27. (In French). https://www.mapaq.gouv.qc.ca/SiteCollectionDocuments/Regions/Mauricie/Uncoursdeau_Cestvivant.pdf

**REPLY BY THE AUTHORS**

The authors sincerely thank Dr. Khebizi H. for the valuable and constructive comments regarding the integration of geomorphological and sediment transport dynamics in flood risk assessments. We fully recognize the importance of these elements in broader flood risk studies. However, we would like to clarify the primary objective and scope of our current work.

In fact, the current study was specifically aimed at evaluating the effectiveness of the existing flood protection measures along Wadi Deffa, with a particular emphasis on assessing the channel's capacity to manage extreme flood events, such as the one that occurred on October 1, 2011. The primary goal was to replicate this event using hydrological and hydraulic modeling, allowing us to analyze the performance of the protective infrastructure. Given this focus on the capacity related to a specific historical flood event, an evaluation of changes due to sedimentation was not considered critical for achieving the study's objectives.

Although the physical characteristics and other attributes of the study region were integral to the hydrologic and hydraulic analyses, it's important to recognize that morphological changes caused by sediment transport, particularly at large scales in impacted areas,

typically occur slowly and gradually. These changes can take a significant amount of time before they visibly alter the landscape. Understanding this gradual transformation is essential for evaluating long-term environmental impacts and formulating sustainable land and water management strategies. While these processes may not yield immediate effects, their eventual repercussions on terrain and watercourses can be considerable, particularly in relation to flood risk assessment, hazard mapping, long-term mitigation strategies, and future urban development planning. Therefore, including these factors in our analysis of short-term structural performance may not significantly improve the immediate assessment of flood protection measures.

While our current study concentrated on immediate flood risk through hydrological and hydraulic modelling, we greatly respect Dr. Khebizi's viewpoint and recognize that future studies, particularly those aimed at urban flood risk management and the development of flood maps, will benefit greatly from these broader considerations.

---

## Author Comment (AC1)

**Rebuttal of Review CC1**

**Dear authors and colleagues of the scientific community,**

**Fist, I would like to thank the authors for the valuable response to my comment concerning Part I and I am pleased to add a second comment for Return period of high-dimensional compound events. Part II: Analysis of spatially-variable precipitation.**

**For this, four questions seems to me interesting to be asked if possible. My first question concerns the RP. It changes in space and time and its occurrence is not necessarily of the same intensity. How can you differentiate short return periods from long-term ones?**

In our approach, the RP is addressed using the *Joint Return Period* (JRP) concept, which considers the simultaneous occurrence of extreme events across multiple locations and various temporal scales. The differentiation between short and long return periods is achieved through the analysis of the joint distribution of extreme events in space and time, utilizing multivariate copulas and Kendall's method to evaluate the critical probability level ($t$). This procedure enables the identification of events with varying frequencies of occurrence, adjusting the results to the spatial and temporal characteristics of the phenomenon.

It is important to highlight that the critical probability level ($t$) depends on the selected return period. In our study, we used a 100-year return period as a reference, but this threshold can be adapted to other values, whether lower or higher, depending on the specific needs of the analysis. The choice of the return period directly influences the value of $t$, which in turn defines the critical hypersurface in the multidimensional space. This hypersurface contains the values of the analyzed variable —in our case, precipitation measured across multiple locations— that are associated with the selected *Joint Return Period* (JRP). Events located on this hypersurface meet the conditions established by the defined return period, allowing their identification and analysis within the applied multivariate framework.

**My second question, in addition to the hydraulic and hydrological study, is it possible to introduce anthropogenic variables, for example, the existence of dams, sewage networks, treatment plants, which can by incidence or overload amplify the risk of flooding?**

We acknowledge the relevance of anthropogenic factors in influencing hydrological processes, particularly in the context of compound events and their potential amplification due to human interventions. While the current study focuses on the statistical modeling of precipitation and its spatial variability, the proposed methodological framework is flexible and allows for the integration of additional variables, such as the presence of dams, sewage networks, and treatment plants, which can significantly alter hydrological responses during compounding events.

The importance of considering such variables has been highlighted in previous

studies, notably by Salvadori et al. (2011), who emphasized the necessity of incorporating system-specific characteristics when defining multivariate return periods and identifying critical design events. In their case study of the Ceppo Morelli dam, the authors modeled key hydrological variables—flood peak discharge, flood volume, and the initial water level in the reservoir—using a trivariate copula-based framework. While the study focused primarily on these hydrological variables, it also recognized the role of structural features, such as storage capacity and spillway levels, in influencing flood behavior.

In this approach, anthropogenic variables can be integrated either directly into the multivariate modeling framework or indirectly by modifying boundary conditions that influence the hydrological response. For example, dam regulation policies, operational constraints, and sedimentation effects can alter flow dynamics, while urban infrastructures, such as drainage networks and treatment plants, can affect the timing and magnitude of runoff. Incorporating these variables would require adjusting the dependency structures within the copula framework or redefining the critical hypersurface associated with specific joint return periods to reflect altered system behaviors.

Although this study primarily focuses on hydroclimatological variables, future research could expand the framework to systematically include anthropogenic elements. Such an extension would enhance the capacity to simulate complex hydrological systems and improve the characterization of compound events, especially in highly managed or urbanized watersheds.

**My third question concerns the implications of geomorphology and the terrigenous material transported and deposited during the flood. For this, I would like to invite you to read my discussion concerning the Evaluation the Effectiveness Of The Existing Flood Risk Protection Measures Along Wadi Deffa In El-Bayadh City, Algeria By Ben Said M., Hafnaoui M.A., Hachemi A., Madi M., Benmalek A. In this discussion, I highlighted the implications of geomorphology and the sedimentary material transported and then deposited during the flood. These are two related factors that change over time where we can follow the evolution of the morphology of the river and quantify the terrigenous material. Using your approach, can you combine runoff morphology and sediment supply in a flood scenario?**

Indeed, fluvial dynamics and sediment transport are fundamental processes that shape river basins and influence the behavior and impacts of compound flood events. The redistribution of terrigenous material during floods alters channel morphology, affects flow patterns, and changes floodplain connectivity, all of which can significantly influence the evolution of future flood events.

While our current study focuses on the statistical modeling of compound precipitation events, the proposed methodological framework—based on multivariate copulas and the identification of critical hypersurfaces—offers the flexibility to incorporate additional variables, such as those related to runoff morphology and sediment transport. This integration could enable the simulation of more comprehensive scenarios that account not only for precipitation magnitude but also for geomorphological processes that affect flood dynamics.

We recognize, however, that incorporating sediment dynamics into a multivariate return period framework presents certain challenges. Key limitations include the scarcity and irregularity of sediment transport data, the complex and often nonlinear relationships between hydrological variables and sediment dynamics, and the difficulty in defining extreme geomorphological events in a way that aligns with hydrological thresholds. Additionally, the dynamic nature of sediment transport and channel morphology, which evolves during flood events, complicates the use of traditional copula models that typically assume static dependencies.

Despite these challenges, the inclusion of geomorphological variables is both feasible and valuable. Future research could explore the use of dynamic copula models or vine copulas to better capture evolving dependencies during flood events. Additionally, coupling the copula-based statistical framework with geomorphological models—such as sediment transport or erosion-deposition models—could provide a more holistic understanding of compound flood events.

Such an integrated approach would offer a more realistic representation of flood scenarios, particularly in regions where sediment dynamics significantly influence flood hazards. It would also enhance the capacity to evaluate the effectiveness of flood risk mitigation measures, especially in sediment-sensitive environments, as highlighted in your discussion on the Evaluation of the Effectiveness of the Existing Flood Risk Protection Measures Along Wadi Deffa in El-Bayadh City, Algeria.

**A final question concerns the lithological vulnerability, particularly erosion and the implication of flooding on urban areas. Is it possible to add variables indicating the lithological vulnerability in the modelling, or should the modelling in your approach be limited to hydroclimatological data?**

**Here attached, my discussion Of Evaluating The Effectiveness Of The Existing Flood Risk Protection Measures Along Wadi Deffa In El-Bayadh City, Algeria By Ben Said M., Hafnaoui M.A., Hachemi A., Madi M., Benmalek A.**

While our current approach focuses on hydroclimatological data, the framework can be adapted to include variables such as soil type, rock composition, and erosion susceptibility to better capture the interactions between geology and flood dynamics.

We acknowledge that integrating lithological data presents challenges due to spatial variability and data availability. However, exploring this integration could provide valuable insights, especially in regions where erosion significantly amplifies flood risks.

Future research could focus on combining statistical models with geotechnical and geomorphological analyses to assess how lithological factors influence flood behavior and urban vulnerability. Your reference to the study on Wadi Deffa in El-Bayadh City, Algeria highlights the relevance of this approach and the potential for more comprehensive flood risk assessments.

---

## Author Comment (AC2)

**Rebuttal of Review 1**

**Journal:** HESS
**Paper:** hess-2024-335
**Title:** Return period of high-dimensional compound events. Part II: Analysis of spatially-variable precipitation
**Author(s):** Diego Urrea Méndez, Manuel Del Jesus, and Dina Vanessa Gomez Rave

**GENERAL COMMENT**

For the reasons given below, I recommend a REJECTION (WITH POSSIBLE RESUBMISSION).

**SPECIFIC COMMENTS**

**Line(s) 26–27**

**AUTHOR(s): Depending on the hydrological scale used, compound precipitation events measured at different rain gauges may contain a considerable number of zeros (zero-inflated data).**

**REFEREE: From a Statistical point of view, it is a nightmare, and the problem is far from being resolved. The introduction of a mixed model, and then considering only positive rainfall values, does not seem to fix the question, for introduces other problems (see my comment below).**

We acknowledge the complexity of handling zero-inflated precipitation data, and we agree that this remains an open challenge in statistical hydrology. Our approach, rather than attempting to definitely "solve" this issue, involves the introduction of a mixed model that explicitly includes both a discrete component for zeros and a continuous component for positive precipitation values (Section 2.1.1). This framework provides a pragmatic solution to mitigate the impact of an excess of zeros on the dependence structure while preserving the intermittent nature of rainfall and systematically capturing spatial dependencies.

Although this method has limitations, it aligns with established practices in hydrology and has been employed in previous studies addressing similar issues (Serinaldi 2008). While not a final solution, it offers a mathematically sound and physically consistent representation of the variability of precipitation.

**Line(s) 36–38**

**AUTHOR(s). One of the first mixed models applied in a bivariate approach was developed by (Shimizu, 1993). This approach represents a copula-based mixed distribution function composed of a continuous part (observations greater than zero) and a discrete part (observations at zero).**

**REFEREE. The model by Shimizu (1993) is not copula-based: the modeling via copulas is present nowhere in the paper, also considering that it deals with a mixture of discrete-continuous distributions. Incidentally, none of the marginals used in that paper is heavy-tailed, possibly inadequate to deal with rainfall extremes.**

Our reference to Shimizu (1993) incorrectly suggested that the model was copula-based. The misstatement stemmed from imprecise wording rather than a misunderstanding of the model. Our intention was to highlight its multivariate nature and the discrete–continuous mixture, not to imply the use of copulas.

Although Shimizu's original model did not address tail behavior, his conceptual framework—partitioning precipitation into discrete and continuous components—laid the groundwork for more advanced approaches, including the one adopted in this study. He underscored the necessity of explicitly handling the intermittent nature of precipitation, an insight that has continually shaped methodological advancements.

We have addressed these aspects in the revised manuscript, as follows:

*"One of the first approaches to separate rainfall into discrete (zero) and continuous (positive) components was the bivariate model proposed by Shimizu (1993). Despite its bivariate focus, it laid a conceptual foundation for handling the intermittency of precipitation, paving the way for future extensions to multivariate contexts with the capacity to model extreme behavior."*

**Line(s) 43–44**

**AUTHOR(s).** Another fundamental definition when discussing floods corresponds to the notion of the return period (RP). Salvadori et al. (2011) defines the RP as the time elapsed between two successive occurrences of a prescribed event.

**REFEREE. The true novelty of the paper by Salvadori et al. (2011), beyond the mathematical formalization of a multivariate notion of RP, and the introduction of original multivariate design techniques, is that the calculation of the RP is written in terms of copulas only in any multi-dimensional setting (not only bivariate).**

As the referee notes, the formulation of the return period (RP) entirely in terms of copulas, as presented by G. Salvadori, C. De Michele, and F. Durante (2011), is a significant theoretical advancement, extending their applicability beyond the bivariate case. However, its practical implementation is constrained by high computational demands. Their study illustrates a three-dimensional case where the computation required approximately 48 hours of CPU time on an iMac with an Intel Core 2 Duo 3.06 GHz processor and 8 GB of RAM, underscoring the prohibitive costs of scaling to higher dimensions. The authors explicitly acknowledge these limitations, emphasizing the need for further research to develop alternative design strategies and refine the theoretical framework for multivariate risk assessment.

In response, our approach enhances computational efficiency, enabling the extension of the analysis to high dimensions without incurring excessive computational costs. This refinement makes multivariate RP calculations not only more feasible but also more practical for real-world applications where complexity demands efficiency.

We propose to reformulate the original text as follows:

*"While Salvadori et al. (2011) provide a solid theoretical foundation for calculating the multivariate return period using copulas in n-dimensional contexts, its practical application faces significant challenges related to computational load, especially in high-dimensional scenarios. In our study, we address this limitation by employing optimized computational techniques that enable the extension of the analysis to five dimensions, reducing computation times and facilitating its application to more complex real-world cases."*

**Line(s) 56–57**

**AUTHOR(s).** In this context, this study has two main objectives: (I) to expand the methodological framework for modeling data with zero intermittency from a bivariate (Shimizu, 1993; Serinaldi, 2008; Villarini et al., 2008), to a five-dimensional approach ...

**REFEREE. Is 5 a magic number, in hydrology or elsewhere? Increasing the dimension is not a synonymous of novelty: what about if tomorrow I publish a paper on a 6-dimensional approach? It may solve problems in 6 dimensions, but may not change the paradigms. . . in addition, you only dealt with a sub-class (Group 32) of the 5 dimensional problem ...**

The selection of five dimensions is not arbitrary but is grounded in the results of the exploratory dependency analysis and the availability of data in the studied region. The strongest dependency structure we identified, based on Kendall's correlation coefficient, corresponded to five stations distributed across different sectors of the watershed. However, this does not imply that the methodology is not scalable to more dimensions.

The added value lies not in merely increasing the dimensionality but in adapting and validating an approach that integrates copulas and mixed models (discrete–continuous) within a multidimensional framework beyond the traditional bivariate or trivariate settings. Therefore, we do not propose "five" as a paradigm but rather demonstrate that the method is scalable to higher dimensions. In addressing the first objective, we acknowledge an overstatement in our wording. Our intent is not to "expand" the framework for modeling data with zero intermittency (Shimizu 1993; Serinaldi 2008; Villarini, Serinaldi, and Krajewski 2008) but rather to adapt it to n-dimensional spaces.

Regarding the apparent 'sub-class' (Group 32), this group was analyzed in detail because the hypersurface used to estimate the 100-year Joint Return Period (JRP) lies fully within this group. This means that the critical level $t$ associated with the 100-year return level is reached only by events within Group 32. While all groups contribute to the cumulative probability leading up to this threshold, none of the events included in them individually reaches the probability level required to belong to this JRP (except some in group 32). Indeed, the probability gradually accumulates across groups until it meets the threshold within Group 32.

However, the approach considers all possible combinations (32 groups) of zeros and positive values across the five stations. In lines 189–192, 206–209, 225–227, 256–260, and 291–293, we present the results for all groups, covering the full range of configurations. As specified, this analysis encompasses the complete set of precipitation scenarios introduced in Equation (1). To improve clarity, we propose adjusting line 56 as follows:

*"In this context, this study has two main objectives: (I) to adapt a methodological framework for modeling data with zero intermittency in an n-dimensional space (Shimizu, 1993; Serinaldi, 2008; Villarini et al., 2008), going beyond traditional bivariate approaches."*

**Line(s) 72–73**

**AUTHOR(s).** This consideration leads to the incorporation of multivariate mixed models, which will be detailed in this chapter.

**REFEREE.** The mixed model ignores/spoils the correlation structure of the sequences of $(0, > 0)$'s in the rainfall time series, and in turn the COMPOUND nature/feature of the events. For example, the total precipitation in the two sequences A and B below is the same (0 means no rain, 1 means rain), but the COMPOUND impact of series B could be devastating as compared to the one of series A:

**A=[0,1,0,1,0,1,0,1,0,1]**

**B=[1,1,1,1,1,0,0,0,0,0]**

**Here, the correct approach would be to use a stochastic renewal process, as in G. Salvadori and C. De Michele. Statistical characterization of temporal structure of storms. Advances in Water Resources, 29(6):827–842, 2006. doi: 10.1016/j.advwatres.2005.07.013**

Our approach is designed to capture the most extreme event occurring at a given time while accounting for spatial dependence. The objective is to characterize the joint dependence of precipitation across multiple locations using the JRP. The model presented does not explicitly account for the sequential temporal structure of precipitation events, as highlighted in the reviewer's example (sequences A and B). Instead, it focuses on the spatial co-occurrence of extreme precipitation. Future research could explore the integration of stochastic renewal processes with multivariate spatial dependence models to capture both the temporal and spatial structure of compound events.

**Line(s) 77**

**AUTHOR(s). 4. Hazard scenario**

**REFEREE. A reference to Salvadori, G., Durante, F., De Michele, C., Bernardi, M., and Petrella, L.: A Multivariate CopulaBased Framework for Dealing with Hazard Scenarios and Failure Probabilities, Water Resources Research, 52, 3701–3721, https://onlinelibrary.wiley.com/doi/abs/10.1002/2015WR017225, 2016 should be put here: it is the first paper where Hazard Scenarios are first formalized in terms of Copulas, including those mentioned by the Authors.**

Following your recommendation, we propose citing this reference in Section 3.4 (*Hazard Scenarios*) to emphasize its pioneering role and further strengthen the theoretical framework of our analysis:

*"3.4 Hazard Scenarios*

*The mathematical definitions for the JRP and critical layer in d-dimensional spaces were developed using the methodology described by Manuel del Jesus et al. (2023). In this study, we focused on identifying multivariate design events with a 100-year RP, associated with the Kendall hazard scenario for the proposed approaches.*

*This work builds upon the theoretical foundations established by Salvadori et al. (2011, 2016), who first formalized hazard scenarios in terms of copulas and introduced the Kendall approach for evaluating multivariate risk scenarios."*

**Line(s) 80**

**AUTHOR(s). Specifically, we compute the critical surface...**

**REFEREE. Surface or hyper-surface, with dimension larger than 2?**

The appropriate term in our study is *hyper-surface*, as the analysis takes place in multidimensional spaces with more than two dimensions. Referring to it as a *surface* was an imprecise simplification that may have led to confusion.

To correct this issue, we will revise the manuscript by replacing *critical surface* with *critical hyper-surface* in the relevant sections.

**Line(s) 94–96**

**AUTHOR(s). To ensure regional representativeness, "regional events" are then selected based on simultaneous rainfall contribution across all stations and, for non-independent events, the highest total precipitation.**

**REFEREE. This sentence is quite obscure: what do you mean by "for non-independent events"? What are the events you are considering? Monthly maxima at different stations? What kind of independence do you consider? Spatial-Pairwise? Spatial-Global? Note that they are different: Pairwise independence may not imply Global one... How do you test it? How do you identify a "homogenous" region? Via clustering procedures?**

*Non-independent events:* In our study, the term "non-independent events" refers to the temporal overlap of events due to the selection method of maximum precipitation events at each station, as described in lines 94-96 of the manuscript. To avoid this overlap, an additional criterion is applied based on total accumulated precipitation, as detailed in Chapter 2.1.

*Events analyzed:* Instead of considering only individual monthly maxima, compound events were identified based on their spatial dependence and hydrologically validated using flow series, selecting those that generated streamflow events exceeding the 10-year RP threshold. The full event selection process is described in Chapter 2.1 and summarized in Figure 4, where the criteria applied for identification and regional representativeness are detailed.

*Spatial dependence:* Our study does not aim to demonstrate independence between stations but rather to identify dependence in the occurrence of precipitation events. To this end, we assess spatial dependence using the Kendall correlation coefficient and model the dependence structure through multivariate copulas (lines 206-209). We did not perform formal spatial independence tests, as our objective is to characterize the dependency relationships between stations rather than to verify their independence.

*Temporal independence:* When referring to independence, we specifically mean the temporal independence of the selected events within each station's time series. To assess this, we apply an autocorrelation analysis and evaluate independence between consecutive events in each time series using the Kendall correlation coefficient (lines 189-192). The results confirm that the selected events do not exhibit significant temporal dependence, as shown in Figure 5 for Group 32, while line 189 provides details for the other groups.

*Homogeneous region:* No formal clustering procedure was applied. Instead, homogeneity was established based on the spatial correlation of precipitation and the consistency in station responses (lines 291-293). Additionally, all selected stations are located within the same watershed, ensuring that they share similar hydrological forcings, thereby justifying their selection without requiring further segmentation.

**Line(s) 105 & Eq. (1)**

**AUTHOR(s). In this context, we have extended the model proposed by Serinaldi (2008)...**

**REFEREE. I do not think it is an extension, except perhaps for the dimension, but then it would be trivial: you simply try to account for the probability of mutually exclusive events in dimensions larger than 2, nothing too special that could justify a specific paper. . . Φ is not precisely defined, what should it represent? A joint CDF? Eq. (1) looks like a linear combination of probabilities: what is its meaning? What is the domain/support of the parameters p's, and their inter-relationships? No information/explanation is provided. . . 2Furthermore, written in this way, the formula may account twice for the probability of zero rainfall: in fact, by definition, F (x) = P (X ≤ x) (in whatever form you write it, for one or more variables) includes the case that the variable(s) take on the value 0, even if the threshold x is strictly larger than 0: the probabilities in Eq. (1) look more like conditional ones. In addition, P0 should depend upon the location, I would be surprised if it were the same at all stations. In all cases, you should prove that Φ is a genuine probability distribution, which however suffers from over-parametrization (i.e., the number of p's): estimating all these parameters in a highdimensional space is a torment, at least from a numerical standpoint, for the estimates almost certainly never correspond to optimal values (at best, they are suboptimal in the most favorable cases).**

As mentioned in our response to lines 56–57, our work does not increase only the dimensionality of the problem, it develops a methodological framework for the computation of the JPR in high dimensions. This endeavor requires addressing challenges related to probabilistic consistency and numerical stability, ensuring that the integration of copulas and mixed models (discrete–continuous) correctly represents the dependence between stations, without double-counting probabilities and without introducing biases in the estimation of joint extreme events.

Regarding the definition of the function Φ in Equation (1), as stated in lines 105–106 of the manuscript, it represents the joint cumulative distribution function (JCDF) of precipitation across the considered stations. This

function is structured within a mixed continuous-discrete model that explicitly accounts for zero intermittency. Equation (1) is not intended as a simple linear combination of probabilities but rather as a representation of the JCDF, where the complete set of events selected is decomposed into homogeneous groups. Equation (1) is the reconstruction of the JCDF over all the events, combining the fits obtained for each one of the groups.

The $p$ parameters represent the probabilities that any given event belongs to each one of the groups, therefore these values are restricted to the interval [0,1]. They sum up to 1, since their combination results in the original set of selected events. This way, $\Phi$ constitutes a convex combination of CDFs, resulting in a CDF itself.

We understand the reviewer's concern regarding the potential duplication of the probability of zeros in Equation (1). We have carefully reviewed this point and clarify that the model's structure is designed to avoid such duplication. The probability $P_0$ represents the joint probability of no precipitation occurring at any station and is calculated independently of the conditional probabilities considered in the subsequent terms of the equation. Moreover, we agree with the reviewer that $P_0$ could vary across stations due to local climatological differences.

As commented above, the values of the $p$ parameter results from the decomposition of events into groups, and thus they do not need to estimated, but rather directly calculated from the proportion of events from the original sample that belong to each one of the groups.

We will revise the manuscript to clarify these aspects and have expanded the explanations related to Equation (1), including an explicit definition of $\Phi$, the domain, and the interrelationships of the $p$ parameters.

**Line(s) 125–126**

**AUTHOR(s). Gaussian Copula without intermittency (Gaussian): This approach considers the joint dependence between compound rainfall events without accounting for zero intermittency, including all rainfall data without exception.**

**REFEREE. The presence of 0's yields Ties, which adversely affect (spoil) statistical techniques: how do you manage such a problem, given the fact that no effective solutions are present in Literature? The failure of the Gaussian approach may be due to the fact that, as remarked below, it is inadequate for dealing with extremes, but also to the fact that Ties play against it in this approach: you must make things clear.**

The selection of the Gaussian Copula was justified in the manuscript, as stated in lines 143–146. It was not chosen for its ability to model extreme events, as its limitations have been widely documented (Jaser and Min 2021), but rather for its capacity to handle high-dimensional data. In this specific approach, ties were intentionally not considered, as the objective was to evaluate how the Gaussian Copula models dependence without distinguishing between zero and positive values and how this affects the results obtained.

In Section 3.2.2 (Pre-treatment of Data, Part I), we detail the limitations that ties introduce in fitting procedures and the biases they generate in multivariate analysis. It is emphasized that ties reduce statistical efficiency and can distort the underlying dependency structure, particularly in rank-based copula models (De Michele et al. 2013; Pappadà, Durante, and Salvadori 2017). To mitigate this issue in approaches that account for intermittency (Gaussian Groups, Vine Gaussian, Vine Extreme, and Vine t-Student) (lines 127-139), we segmented the data into groups based on the presence or absence of zero values and applied copulas only to strictly positive data within each group. This methodology eliminates the adverse effects of ties, enabling a more accurate estimation of the dependency structure.

**Line(s) 128–129**

**AUTHOR(s). This method models each group using the Gaussian copula, leveraging its ability to model high dimensions.**

**REFEREE. Unfortunately, a Gaussian framework is unsuitable for dealing with maxima, such as those considered in this paper. .**

This limitation was acknowledged in the manuscript, where we clarified that the selection of the Gaussian Copula was not based on its ability to model extremes. As mentioned in lines 143–145, this clarification had already been addressed in the response to Lines 125–126.

The purpose of including this approach was to highlight its limitations in this context. Since your comment is closely related to previous observations regarding the Gaussian Copula, we refer to that response to avoid redundancies and maintain consistency in the discussion.

**Line(s) 130–132**

**AUTHOR(s). This approach utilizes R-vine structures with Gaussian copulas to model all pairs of series. It combines the flexibility of R-vines for capturing complex dependence structures with**

the efficiency of Gaussian copulas for pairwise modeling.

**REFEREE. I am not sure that a Gaussian copula could be "efficient" (whatever you mean with this unspecified feature) if the true dependence structure is not Gaussian itself. A Gaussian copula has feasible mathematical properties, but also strong limitations, especially considering Extreme phenomena, as abundantly pointed out in Literature.**

We agree that the Gaussian copula has significant limitations, as previously mentioned. Its use in this context is justified by its computational simplicity within R-vine structures and its ability to handle high dimensionality, rather than its suitability for modeling extremes. This clarification will be incorporated into the manuscript to avoid any potential misunderstandings.

*"Vine Gaussian copulas (Vine Gaussian): This approach utilizes R-vine structures with Gaussian copulas to model all pairs of series. While Gaussian copulas offer computational simplicity for pairwise modeling in high-dimensional spaces, they are limited in capturing tail dependencies, making them less suitable for extreme event analysis (Jaser and Min, 2021). In this study, their use serves as a baseline for comparison with more flexible copula families."*

**Line(s) 133**

**AUTHOR(s). Vine extreme copulas (Vine extreme): This approach uses R-vine structures with a diverse set of bivariate copulas...**

**REFEREE. Perhaps, dealing with maxima, Extreme Value copulas should better be used, but these exclude the case of negative dependencies: a justification is required here.**

As stated in the manuscript (line 133), the Vine extreme approach explicitly accounts for the possibility of negative dependencies by incorporating rotated versions of Archimedean copulas. This inclusion directly addresses the limitations raised by the reviewer, ensuring that the model can capture a wide range of dependency structures, including those involving negative relationships when necessary.

Additionally, within the Vine extreme structure, bivariate dependencies in extreme events are modeled using Extreme Value copulas, specifically Gumbel and Joe, ensuring an adequate representation of strong tail dependencies. The use of R-vine structures allows for the selection of the most appropriate copula for each pair of variables based on their dependency relationship, providing greater flexibility and enhancing the model's adaptability to complex hydrological scenarios.

**Line(s) 143–146**

**AUTHOR(s). The selection of the Gaussian copula [...] is supported by its frequent application in climate and hydrological research focused on simulating extreme conditions (Chen and Guo, 2019).**

**REFEREE. This sentence/explanation sounds like a suicide, for it reads as: since (inexperienced) practitioners frequently use the Gaussian copula, this justifies its use, and therefore we use it. No comment.**

The selection of the Gaussian copula was never presented as a validation of its suitability for modeling extremes. Its well-documented limitations in capturing tail dependencies are precisely why it was included—to illustrate the consequences of its application in this context. By incorporating it as a benchmark, we reaffirm how its use in extreme event analysis can lead to misinterpretations or suboptimal conclusions (Jaser and Min 2021).

At first glance, this might seem futile—Why emphasize what is already well established?—. However, as stated in the response to Lines 143-145, its use in such contexts persists in the literature (as shown by recent studies (García et al. 2021; Mascolo et al. 2024)). If emphasizing these limitations seems redundant, it is only because the persistence of its use suggests that the message has yet to fully resonate. Our study provides further evidence of these limitations and demonstrates how the presence of zero intermittency and ties exacerbates the challenges associated with its use in this type of analysis.

To ensure that the intent of the analysis is accurately conveyed, we propose the following revision:

*"The selection of the Gaussian copula was not intended to justify its use for modeling extremes, given its known limitations in capturing tail dependencies (Jaser and Min, 2021). Instead, it was deliberately included to highlight the consequences of its application in extreme event analysis and to serve as a baseline for comparison with more suitable copulas, as evidenced by the results."*

**Line(s) 149–150**

**AUTHOR(s). To carry out our analysis comprehensively, we have selected 5 strategically distributed rain gauge stations...**

**REFEREE. What do you mean by "strategic"? Do you mean "representative" of the hydrological regime (whatever the word "representative" could mean)? What regionalization/clustering procedures/criteria did you use to decide that these are "strategic"? Or these stations are the only ones available (and so the number 5 has a justification)?**

The term *"strategic"* may have lacked clarity. To avoid misinterpretation, the manuscript has been revised to explicitly outline the criteria used for station selection. As mentioned in response to Lines 56-57, the five rain gauges were chosen not arbitrarily but based on an exploratory dependency analysis and the availability of consistent data within the study region. This implies that, while formal regionalization or clustering techniques were not applied (as noted in the response to Lines 94-96), the selection was nonetheless informed by exploratory analysis, ensuring that the chosen stations appropriately represent the hydrological variability of the basin.

To ensure greater precision in the manuscript, we propose the following revision, which clarifies that the selection was guided by the dependency structure and required spatial coverage rather than being random or based solely on data availability:

*"To carry out our analysis comprehensively, we selected five rain gauge stations based on the results of an exploratory dependency analysis and the availability of continuous, high-quality historical data. The selection was not arbitrary; instead, it was driven by the need to capture the spatial and temporal variability of precipitation events within the basin. The five stations, located across different sectors of the basin, represent the optimal structure of spatial dependence identified during the analysis."*

**Line(s) 155–156**

**AUTHOR(s). A rigorous quality control process was implemented, including outlier identification (Gonzalez and Bech, 2017), review of repeated values...**

**REFEREE. What do you mean by "review of repeated values"? And rigorous with respect to what benchmarks?**

By *"review of repeated values"*, we refer to the identification of unusual sequences of identical values in daily precipitation records, which could indicate systematic measurement errors or sensor malfunctions. While consecutive days with zero precipitation are expected, extended sequences of identical positive values may suggest issues in the data recording and require verification.

Regarding the term *"rigorous"*, while its nuance may seem ambiguous when translated from Spanish to English, it specifically refers to the quality control process adhered to established methodologies and recognized hydrological data management standards (Gonzalez and Bech 2017; Llabrés-Brustenga et al. 2019). To provide a clearer explanation, the quality control process included several key stages to ensure the reliability and consistency of the data. First, outlier detection was carried out using standard statistical techniques to identify potential anomalies in precipitation records and physically impossible values. Subsequently, the verification of null data and false zeros was performed following the criteria established by Lez-Rouco (2001), ensuring that days without precipitation were accurately represented and that there were no erroneous data gaps. Finally, a manual review of extreme events was conducted in cases where automatic algorithms detected discrepancies, providing an additional layer of validation and ensuring the accuracy of the data.

To enhance readability, we propose revising the text as follows:

*"A quality control process was implemented, following established hydrological data management standards (Gonzalez and Bech, 2017; Llabrés-Brustenga et al., 2019). This process included the identification of outliers using standard statistical techniques. It also involved the verification of null values and false zeros, following the criteria outlined by Lez-Rouco (2001), ensuring that days with no recorded precipitation were accurately represented and that potential data gaps were addressed. Additionally, a manual review of extreme events was conducted in cases where discrepancies were detected by automated algorithms, providing an additional layer of validation to maintain data integrity. This multi-step approach ensured that the dataset used in the analysis was both reliable and consistent, meeting rigorous quality control benchmarks."*

**Line(s) 175–177**

**AUTHOR(s). Given the considerable number of groups and to simplify the interpretation of the findings, we will focus on the group where rainfall occurs simultaneously in all stations (group 32 - Fig. 1).**

**REFEREE. So what? You introduce a tricky model, then you realize it is too complex, and thus you use the simplest case given by Group 32: essentially, it corresponds to a classical "AND" hazard scenario. The fact that the model is a mathematical mess was already clear in Eq. (1), so why not considering the case of Group 32 directly, which simplifies the discussion, as well as**

**the mathematical treatment. In practice, you boasted about solving a problem in 5 dimensions, but then you only dealt with a specific sub-case.**

The model considers all possible combinations of precipitation occurrence across the stations, structured into 32 groups, each representing a distinct precipitation pattern (Figure 1). This decomposition avoids part of the complications related to zero precipitation by separating different event configurations. Some examples include:

- Group 1 = [1,0,0,0,0] → Includes events where precipitation occurs only at the last station.

- Group 2 = [1,1,0,0,0] → Includes events where precipitation occurs at the last two stations.

- Group 3 = [1,1,1,0,0] → Includes events where precipitation occurs at the last three stations.

- ...

- Group 32 = [1,1,1,1,1] → Includes events where precipitation occurs at all stations.

Group 32 was analyzed in detail because the hypersurface used for the estimation of the 100-year JRP lies within this group, as previously explained in our response to Lines 56-57. However, this does not mean that the analysis was limited to this case. All groups were modeled and incorporated into the construction of the JCDF presented in Equation (1), which fully accounts for the multivariate dependence structure.

To avoid misinterpretations, we will revise the manuscript to explicitly clarify that the model incorporates all precipitation occurrence scenarios, while the focus on Group 32 is justified by its role in defining the RP.

**Line(s) 185–187**

**AUTHOR(s). Figure 5 presents the autocorrelation plots calculated for group 32. When analyzing the autocorrelation plot of the five event series, it is observed that there is no significant correlation between values at different time intervals.**

**REFEREE. To the best of my understanding of the plot, quite a few estimates of the ACF are outside a (traditional) 5% Confidence Band, and thus I would suspect that the data ARE auto-correlated. . .**

The fact that some values fall outside the confidence intervals of the autocorrelation function (ACF) does not necessarily mean that they indicate significant autocorrelation. As Box et al. (2015) noted, even in non-autocorrelated data, approximately 5% of autocorrelation coefficients are expected to exceed the confidence bands purely due to random fluctuations. Therefore, relying solely on visual inspection can be misleading. A statistical assessment is necessary to determine whether these deviations reflect genuine autocorrelation or are merely artifacts of chance.

To ensure a more rigorous evaluation, we propose incorporating the Ljung-Box test (Ljung and Box 1978) to statistically assess autocorrelation across all lags. This will provide a more conclusive determination of whether the observed deviations result from random variation or indicate structural autocorrelation. The results of this test will be included in the revised manuscript to strengthen the analysis and offer a clearer, statistically grounded interpretation of the autocorrelation patterns in the data.

**Line(s) 195**

**AUTHOR(s). In the upper triangular matrix, Kendall's values are displayed in a heatmap...**

**REFEREE. Confidence Intervals for the estimates must also be provided.**

The addition of confidence intervals for Kendall's $\tau$ estimates is a valuable suggestion. While the manuscript currently presents the point estimates of Kendall's $\tau$ in the heatmap, illustrating the strength and direction of dependencies between station pairs, we understand that incorporating confidence intervals will help quantify the uncertainty associated with these estimates and offer a more robust interpretation of the observed correlations.

In response, we will incorporate confidence intervals for the Kendall's $\tau$ estimates using suitable methods. These intervals will be incorporated into the heatmap and further discussed in the manuscript, ensuring a more comprehensive interpretation of the dependencies. The updated results will be applied to all relevant groups and reported accordingly.

**Line(s) 203–204**

**AUTHOR(s). The results from this indicator showed that both upper and lower tail dependence were present in the data.**

**REFEREE. Believe me, with such data you cannot really claim anything about the possible (statistical) presence of Tail Dependence: this is just visual statistics, too often a deceiving practice used by inexperienced practitioners...**

We understand that relying solely on visual methods for identifying tail dependence may raise concerns about the validity of the conclusions. To clarify, the identification of tail dependence in our study was not based solely on visual inspection. We employed the non-parametric estimator by Schmidt and Stadtmuller (2006), which is specifically designed to detect tail dependencies in multivariate contexts. This approach, as stated in the manuscript (lines 203–205), is more robust and statistically sound than visual methods.

We are fully aware of the limitations of this method, as discussed in the manuscript, and interpreted the results with caution, also considering previous evidence on the occurrence of tail dependencies in extreme precipitation events (Serinaldi 2008; Evin, Favre, and Hingray 2018).

To address your comment and eliminate any ambiguity, we propose the following revision in the manuscript:

*"As in other studies (Brunner et al., 2018), the estimator by Schmidt and Stadtmüller (2006) was applied to determine tail dependencies, acknowledging the limitations associated with this method (Serinaldi et al., 2015). While the graphical representation aids in visualizing dependencies, the statistical evaluation provided by this estimator ensures that the analysis goes beyond visual inspection. The results indicated both upper and lower tail dependencies, though, as highlighted by Serinaldi (2008) and Evin et al. (2018), upper tail dependence is often expected in extreme precipitation events."*

**Line(s) 226–227**

**AUTHOR(s). Additionally, the QQ plots for each group were checked, and it was observed that the GEV adequately represented the tail behavior.**

**REFEREE. Formal Monte Carlo Goodness-of-Fit tests, and the corresponding p-values, would be less visual and more objective (e.g., Kolmogorov-Smirnov, or even better Anderson-Darling ones).**

While QQ plots were employed in the manuscript to visually assess the fit of the marginal distributions to the GEV (Generalized Extreme Value) distribution, we are aware that this approach, though informative, relies on subjective visual inspection.

In response to your suggestion, we will incorporate formal goodness-of-fit tests, such as the Kolmogorov-Smirnov and Anderson-Darling tests, to more rigorously evaluate the fit, particularly in the tails of the marginal distributions. These tests will provide a more objective and quantitative assessment, allowing for a clearer understanding of the model's performance in capturing the extremes.

**Table 1**

**REFEREE. In Table 1, GoF p-values are missing for the first two cases, they should be shown.**

To ensure completeness and consistency in presenting the results, we will include the missing p-values for these cases in the revised manuscript.

**Line(s) 256–257**

**AUTHOR(s). To analyze the results for the remaining groups, a box plot was constructed, as presented in Fig. 8, where the distributions of AIC for all groups in each proposed approach are compared.**

**REFEREE. You must first check that the model is admissible via a GoF test, and then (and only then) select the "best" model (according to some criterion) ONLY among the admissible ones. The plots of the AIC's alone in Fig. 8 are of little interest/significance: the corresponding models could all be non-admissible without, first, carrying out suitable GoF tests.**

We understand the importance of ensuring model admissibility through Goodness-of-Fit (GoF) tests before comparing models based on criteria such as AIC. To clarify, the models presented in Figure 8 were all evaluated using GoF tests prior to comparison, ensuring that only admissible models were included in the analysis.

The purpose of Figure 8 is to illustrate the differences in AIC values among models that have already passed the GoF tests, allowing for an objective comparison based on their relative efficiency. However, we acknowledge that this validation process may not have been clearly communicated in the manuscript.

To prevent any confusion, we will revise the text to explicitly state that the models in Figure 8 were subjected to GoF tests prior to comparison. Additionally, we will include a discussion of the GoF test results, providing more transparency regarding the model validation and selection process.

**Line(s) 271–272**

**AUTHOR(s). The first was crucial for assessing whether the dependency of the observed values was maintained, reduced, or improved.**

**REFEREE. How you could "improve" a dependence remains a mystery to me. . .**

The term *improved* may have caused some confusion. Our intention was to express that the analysis aimed to determine whether the model accurately captured the dependency structure of the observed data and whether the generated synthetic data preserved this structure. To avoid ambiguity, we have revised the wording in the manuscript and replaced 'improved' with a more precise formulation:

*"The first was crucial for assessing whether the dependency observed in the original data was accurately captured in the synthetic data generated by the model."*

**Line(s) 283–284**

**AUTHOR(s). This finding supports the ability of the copulas used to accurately capture and reproduce the behavior of the real variables in terms of their extremes and dependencies.**

**REFEREE. Statistically speaking, at most you can hope it: your conclusions are only based on visual analyses, be careful.**

While we acknowledge that graphical methods, such as Kernel Density Estimation (KDE) plots, are inherently exploratory, they are valuable tools widely used in the scientific literature to identify complex patterns, such as tail dependencies, especially when analyzing extreme data.

In our approach, KDE plots were not used in isolation but as complementary tools that facilitate the visualization of data concentration in the tails and aid in interpreting the results. However, we agree that the conclusions can be strengthened by integrating quantitative measures that support the visual observations. To enhance the robustness of our conclusions, we propose incorporating the results from the tail dependence estimator by Schmidt and Stadtmuller (2006) or another suitable estimator. This modification will allow for an objective quantification of the presence of tail dependence, thereby combining the interpretative value of visual analyses with the rigor of formal statistical methods.

**Line(s) 291**

**AUTHOR(s). Based on the analyzed results, the Vine extreme approach demonstrated its ability to reproduce upper tail dependencies.**

**REFEREE. You should add: assuming it is really present.**

The suggestion will be incorporated to ensure the statement accurately reflects the presence of tail dependencies. Additionally, to enhance the rigor and consistency of the analysis, we will explicitly support this assertion with statistical tests that quantify tail dependence. This adjustment will reinforce the conclusions by integrating both graphical and numerical evidence, strengthening the robustness of the presented analysis.

**Line(s) 324–327**

**AUTHOR(s). To calculate the critical level t, it was necessary to calculate the 100-year RP. Considering that we have more values per year than in the case of annual maximum, the quantiles in this case move to the extreme part of the distribution. Note also that each Kendall function is calculated from the continuous part of the function described in Eq. (1), that is, it considers the complete CDF.**

**REFEREE. This claim is quite obscure, and should be clarified. Intuitively, it should be enough to properly set the constant in the definition of the Kendall RP to fix the right temporal scale (e.g., = 1/12). However, this looks like a Kendall RP conditional to the fact that rain is present.**

The interpretation of the Kendall return period in this study follows the methodology outlined by G. Salvadori, C. De Michele, and F. Durante (2011). As noted, the temporal scale can be adjusted by appropriately setting the constant $\mu$. In our case, the number of events per year is explicitly incorporated into the computation of the 100-year RP, ensuring that the temporal framework is properly accounted for.

Regarding the conditioning on rainfall occurrence, it is important to clarify that the Kendall function is calculated using the complete distribution described in Equation (1), which includes both the discrete part (probabilities of zero) and the continuous part (positive values). This means that our JRP estimation properly considers precipitation intermittency across different stations, without exclusively conditioning on the presence of rainfall at all stations.

To make this clearer in the manuscript, we propose the following revision:

*"The critical Kendall level t was calculated following the methodology outlined in Salvadori et al. (2011). Additionally, the actual number of events per year was considered when computing the 100-year return period, which leads to a shift of the quantiles toward the extreme part of the distribution, as highlighted in the analysis. It is important to note that we calculated the Kendall function using the complete Join CDF described in Equation (1), which includes both discrete (zero values) and continuous (positive precipitation) components, thus properly accounting for precipitation intermittency across stations without conditioning exclusively on rainfall presence."*

**Line(s) 329–330**

**AUTHOR(s). The best-performing approach (4) obtained a critical value of 0.993, while the lowest performing approach (1) obtained a critical value of 0.778.**

**REFEREE. Assuming that these results make sense, you should interpret them, and discuss the consequences.**

The difference in the critical values obtained reflects the ability of each approach to accurately capture extreme dependencies between variables. The best-performing approach (0.993) more accurately preserves the dependency structure of the data, indicating a greater capacity to model extreme compound events, especially those impacting multiple stations simultaneously. This result is essential in the context of hydrological risk management, as it enables more reliable estimations of low-frequency, high-impact events.

Conversely, the lowest-performing approach (0.778) underrepresents extreme dependence, which may lead to an underestimation of the risk associated with simultaneous extreme events. In hydrological and disaster management contexts, this outcome could lead to insufficient preparedness and an underestimation of actual risks.

To present the information more effectively, we will refine the manuscript as detailed below:

*"The best-performing approach (4) obtained a critical value of 0.993, indicating its strong ability to preserve the dependency structure observed in the data. This enhances the model's reliability in representing compound events and strengthens its predictive capacity. In contrast, the lowest-performing approach (1), with a critical value of 0.778, shows a reduced ability to maintain data dependencies, which could impact the accuracy of simulations and lead to potential misinterpretations in risk assessments"*

**Line(s) 341–344**

**AUTHOR(s). This procedure was iterated until we obtained a sufficient number of events on the critical layer for each approach. Iteration was necessary because, given the specific nature of the critical level t, only a small fraction of the synthetic events would correspond exactly to this value.**

**REFEREE. Frankly speaking, I really doubt that any of the events generated actually lies on the critical layer, if only for the sake of numerical approximation. Most likely, you have fixed some tolerance coefficient: you must clearly explain how you accept that an event lies on the critical layer.**

The procedure for selecting events on the critical layer is detailed in the manuscript (Line 336, Section 3.5.1 *Critical Layer*). We note that our previous wording may not have been entirely precise. Given the numerical constraints of the process, it is highly unlikely that a randomly generated event would match the critical level $t$ exactly: To handle this numerical limitation, a tolerance coefficient of $10^{-4}$ was applied, allowing events within this range to be accepted during the filtering process.

Since this study operates in a five-dimensional space, ensuring statistical robustness required generating a sufficiently large and representative set of synthetic events. Gaussian Process Regression (GPR) models were employed to capture the structure of the high-dimensional space, followed by an iterative filtering process to extract events meeting the tolerance criterion, resulting in a final set of 1 million critical events.

We propose the following revision in the manuscript:

*"For each approach, a sufficiently extensive set of synthetic events was generated to represent possible realizations within the multidimensional space. The generation process, while computationally intensive, was made feasible and efficient through pre-trained Gaussian Process Regression (GPR) models. The joint distribution function was calculated for each synthetic event, and only those that matched the critical level t were selected. Given the inherent numerical constraints, a a tolerance coefficient of $10^{-4}$ was applied to identify events within the defined range. This approach, consistent with the principles outlined by Salvadori et al. (2011), provided a structured and reliable representation of the critical layer".*

**Line(s) 354–355**

**AUTHOR(s). Solving this loss of precision in high dimensions was easy because we had sufficient event combinations on the critical layer. For each combination of events, we calculated the density function and selected the one with the highest density.**

**REFEREE. It is not clear what you mean by a "combination of events", and how it is chosen. What is its sample size and how is it decided? What is its density function (the joint one?) More details must be given for the sake of discussion and reproducibility.**

When we refer to average precipitation, our goal is to estimate the total precipitation over the watershed, rather than simply computing an arithmetic mean. To achieve this, we use different approaches for the univariate and multivariate cases.

In the univariate case, extreme precipitation values are first estimated independently at each station based on a 100-year RP. To obtain the total precipitation over the watershed, these values are spatially aggregated using Thiessen polygons and hydrological reduction factors, commonly used methods for estimating spatial precipitation from point measurements.

In the multivariate case, we account for spatial dependence between stations, which is not considered in the univariate approach. Instead of aggregating independent extremes, the total precipitation over the watershed is estimated by modeling the joint probability structure of extreme events, ensuring that the spatial correlation of precipitation is preserved when assessing extreme conditions.

While a traditional precipitation average may not fully align with extreme value methods, spatially integrating extreme values across the watershed provides a relevant metric for assessing the system's response under critical conditions. This approach ensures consistency within the extreme value analysis framework, maintaining a valid comparison between the univariate and multivariate methods.

We acknowledge that this procedure may require further clarification in the manuscript and therefore propose the following revision:

*"To compare the results of univariate and multivariate analyses, the total precipitation over the watershed was estimated using both approaches. In the univariate case, the 100-year return period values were calculated independently at each station and aggregated using Thiessen polygons and hydrological reduction factors. In contrast, the multivariate approach incorporated spatial dependence among stations, providing an alternative estimation of total precipitation under extreme conditions. This ensures a consistent and meaningful comparison between methodologies, allowing for a better understanding of how spatial dependence influences extreme event estimation over the watershed."*

**Line(s) 371–372**

**AUTHOR(s). To compare the results of univariate and multivariate analysis, it was necessary to calculate the average precipitation in the watershed using both approaches.**

**REFEREE. Average precipitation could have little to do with the Extreme Value approach: I understand that it is part of common hydrological practice, but then it seems that the Authors are playing at the same time on two different layers, as if they were trying to have a foot in both camps. A justification is required here.**

The concern regarding the use of average precipitation in the context of extreme value analysis is well taken. The comparison between univariate and multivariate approaches does not rely on a simple average of daily or monthly precipitation but rather on the spatial average of extreme values estimated at each station, that is, we compute the average spatial rainfall over the complete watershed of the extreme event considered. This way to proceed ensures consistency with extreme value analysis while providing an aggregated metric that facilitates hydrological interpretation.

The spatial average was computed using extreme precipitation values associated with a 100-year RP, estimated independently at each station. In the univariate approach, these values were derived through standard extreme value analysis techniques and then aggregated using spatial reduction methods, such as Thiessen polygons and hydrological reduction factors tailored for large watersheds.

While a traditional precipitation average may not align with extreme value methods, averaging extreme values across the watershed offers a relevant measure of the system's aggregated response under critical conditions. This approach ensures that the comparison remains within the framework of extreme value analysis rather than introducing inconsistencies between methodologies.

We recognize that the explanation of this procedure may require further clarification in the manuscript and therefore propose the following revision:

*"To compare the results of univariate and multivariate analyses, the spatial average of the extreme precipitation values estimated at each rain gauge was used as a common metric. In the univariate approach, the 100-year return period values were calculated independently for each station and aggregated using Thiessen polygons and spatial reduction factors appropriate for large watersheds. This procedure ensures a consistent and meaningful comparison between approaches, enabling the assessment of differences in estimated extreme events across the watershed."*

**Line(s) 379–381**

**AUTHOR(s). Compared to this method, the Gaussian, Gaussian Groups, and Vine Gaussian models tend to underestimate the events, while the Vine t-student overestimates them.**

**REFEREE. Here, as well as below, you cannot speak about under- or over-estimates: this makes sense only if you know the true value. Here you can only speak about relative smaller/larger values.**

We agree that terms such as *"underestimation"* or *"overestimation"* imply a known true value, which does not apply in this context. Our comparison is based on a reference model—the one that best fits the data—against which other models are evaluated. In this framework, the Gaussian, Gaussian Groups, and Vine Gaussian models yield relatively smaller values, while the Vine t-student model produces relatively larger values. These differences are assessed relative to the reference model rather than an absolute benchmark, offering a comparative perspective on how each model represents extreme events and captures the dependency structure in the data.

This distinction will be clarified in the text, and the proposed modification is:

*"Compared to the model that showed the best fit to the data, the Gaussian, Gaussian Groups, and Vine Gaussian models produced relatively smaller values, while the Vine t-student yielded relatively larger values. These comparisons are made relative to the reference model and do not imply absolute under- or over-estimation. Instead, they highlight the differences in how each approach captures the dependence structure and the extremes present in the dataset."*

**Line(s) 399–ff.**

**REFEREE. The Discussion and the Conclusions sections could/should be merged in a single section "Discussion & Conclusions".**

In response to the suggestion, the Discussion and Conclusions sections have been consolidated into a single "Discussion & Conclusions" section. This streamlined format presents key findings, interpretations and limitations, thereby enhancing overall clarity and coherence.

**References**

Box, George E. P., Gwilym M. Jenkins, Gregory C. Reinsel, and Greta M. Ljung. 2015. *Time Series Analysis: Forecasting and Control.* John Wiley & Sons.

De Michele, C., G. Salvadori, R. Vezzoli, and S. Pecora. 2013. "Multivariate Assessment of Droughts: Frequency Analysis and Dynamic Return Period." *Water Resources Research* 49 (10): 6985–94. https://doi.org/10.1002/wrcr.20551.

Evin, Guillaume, Anne-Catherine Favre, and Benoit Hingray. 2018. "Stochastic Generation of Multi-Site Daily Precipitation Focusing on Extreme Events." *Hydrology and Earth System Sciences* 22 (1): 655–72. https://doi.org/10.5194/hess-22-655-2018.

G. Salvadori, C. De Michele, and F. Durante. 2011. "On the Return Period and Design in a Multivariate Framework." *Hydrology and Earth System Sciences* 15 (11): 3293–3305. https://doi.org/10.5194/hess-15-3293-2011.

García, J. Agustín, Mario M. Pizarro, F. Javier Acero, and M. Isabel Parra. 2021. "A Bayesian Hierarchical Spatial Copula Model: An Application to Extreme Temperatures in Extremadura (Spain)." *Atmosphere* 12 (7): 897. https://doi.org/10.3390/atmos12070897.

Gonzalez, Sergi, and Joan Bech. 2017. "Extreme Point Rainfall Temporal Scaling: A Long Term (1805-2014) Regional and Seasonal Analysis in Spain: EXTREME POINT RAINFALL TEMPORAL SCALING IN SPAIN." *International Journal of Climatology* 37 (15): 5068–79. https://doi.org/10.1002/joc.5144.

Jaser, Miriam, and Aleksey Min. 2021. "On Tests for Symmetry and Radial Symmetry of Bivariate Copulas Towards Testing for Ellipticity." *Computational Statistics* 36 (3): 1–26. https://doi.org/10.1007/s00180-020-00994-0.

Lez-Rouco, J Fidel Gonza. 2001. "Quality Control and Homogeneity of Precipitation Data in the Southwest of Europe." *Journal of Climate* 14: 15.

Ljung, G. M., and G. E. P. Box. 1978. "On a Measure of Lack of Fit in Time Series Models." *Biometrika* 65 (2): 297–303. https://doi.org/10.1093/biomet/65.2.297.

Llabrés-Brustenga, Alba, Anna Rius, Raúl Rodríguez-Solà, M. Carmen Casas-Castillo, and Angel Redaño. 2019. "Quality Control Process of the Daily Rainfall Series Available in Catalonia from 1855 to the Present." *Theoretical and Applied Climatology* 137 (3-4): 2715–29. https://doi.org/10.1007/s00704-019-02772-5.

Mascolo, Valeria, Alessandro Lovo, Corentin Herbert, and Freddy Bouchet. 2024. "A Gaussian Framework for Optimal Prediction of Extreme Heat Waves." EGU24-18866. Copernicus Meetings. https://doi.org/10.5194/egusphere-egu24-18866.

Pappadà, R., F. Durante, and G. Salvadori. 2017. "Quantification of the Environmental Structural Risk with Spoiling Ties: Is Randomization Worthwhile?" *Stochastic Environmental Research and Risk Assessment* 31 (10): 2483–97. https://doi.org/10.1007/s00477-016-1357-9.

Schmidt, Rafael, and Ulrich Stadtmuller. 2006. "Non-Parametric Estimation of Tail Dependence." *Scandinavian Journal of Statistics* 33 (2): 307–35. https://doi.org/10.1111/j.1467-9469.2005.00483.x.

Serinaldi, Francesco. 2008. "Analysis of Inter-Gauge Dependence by Kendall's K, Upper Tail Dependence Coefficient, and 2-Copulas with Application to Rainfall Fields." *Stochastic Environmental Research and Risk Assessment* 22 (6): 671–88. https://doi.org/10.1007/s00477-007-0176-4.

Shimizu, Kunio. 1993. "A Bivariate Mixed Lognormal Distribution with an Analysis of Rainfall Data." *Journal of Applied Meteorology* 32 (2): 161–71. https://doi.org/10.1175/1520-0450(1993)032%3C0161:ABMLDW%3E2.0.CO;2.

Villarini, Gabriele, Francesco Serinaldi, and Witold F. Krajewski. 2008. "Modeling Radar-Rainfall Estimation Uncertainties Using Parametric and Non-Parametric Approaches." *Advances in Water Resources* 31 (12): 1674–86. https://doi.org/10.1016/j.advwatres.2008.08.002.

---

## Author Comment (AC3)

**Rebuttal of Review 2**

**The manuscript addresses the analysis of precipitation across spatially-related sites with a focus on modeling zero-inflated data. It places significant emphasis on the multivariate nature of the problem, demonstrated through a five-dimensional application. While the topic is interesting and relevant, the manuscript requires greater detail in several areas to adequately convey the value and robustness of the proposed methodology. Below are specific comments and suggestions for improvement:**

**General CommentsComputational Complexity and Data Requirements:**

**The model estimates $2^d$ distributions, where d is the number of sites. This naturally entails considerable computational costs. Additionally, since each model is estimated on a subset of the data (e.g., Group 32 is only modeled when all variables are non-zero), the approach presupposes access to a large dataset. These aspects warrant discussion and explicit acknowledgment within the manuscript.**

While the estimation of $2^d$ distributions may suggest high computational costs, the parameter estimation process is efficient. Modern statistical packages, such as *VineCopula*, incorporate optimized algorithms that handle this task with minimal computational expense, even in high-dimensional settings. As a result, this step remains computationally feasible without requiring further simplifications. However, the main computational cost arises from estimating the critical hypersurface, which entails a high computational cost. To address this complexity, we employed Gaussian Process Regression (GPR).

Regarding data availability, the need for sufficiently large sample sizes is a well-known challenge in the literature (Brunner 2023; Serinaldi 2013). A widely used strategy to overcome this issue is the generation of synthetic data, which improves the representation of less frequent events while preserving key statistical dependencies. In this study, we applied this approach by fitting multiple copula families to the observed dataset, selecting the best-fitting model based on statistical tests, and generating additional samples accordingly. To ensure that the synthetic data accurately reflected the dependence structure of the original dataset, we compared and validated their correlation coefficients, as discussed in Section 3.3.2.

To strengthen the manuscript, we will explicitly address these aspects, highlighting the computational efficiency of the model and the approach taken to address data limitations.

**Detailed Comments**

**Equation (1):**

**Greater clarity is required for the parameters p0,p1,..., etc, to ensure they constitute a valid probability model. Specifically, do these parameters sum to 1? Providing this information would strengthen the theoretical foundations of the model.**

The parameters $p_0, p_1, ... p_n$ represent the probabilities associated with each group, where each group corresponds to a specific pattern of rainfall across the analyzed stations. Since these probabilities describe mutually exclusive and collectively exhaustive events, their sum is equal to 1, ensuring that Equation (1) defines a valid probability distribution (further details on the group configurations are provided in the *Response to Figure 1*).

To address this point, we will revise the manuscript to clearly articulate this condition, ensuring a more transparent representation of the framework.

**Model Descriptions (Page 5):**

**Model 1 (Gaussian Copula Without Intermittency):**

**Does this model fit a copula to zero-inflated data without accounting for the discrete component of the marginal distributions? If so, is it appropriate to apply copulas to non-continuous data? Addressing this issue would clarify the legitimacy of the approach.**

The model applies a Gaussian copula directly to the data without explicitly accounting for the discrete component associated with zero intermittency in the marginal distributions. This choice was intentional, allowing us to assess the impact of treating the data as fully continuous, particularly in extreme event scenarios.

The use of copulas in non-continuous data with zero inflation can bias dependence estimation by assuming continuity. To address this limitation, the other approaches explored in this study (lines 125-138) adapt the methodology of Serinaldi (2009). However, we intentionally use the Gaussian copula as a reference to illustrate the impact of ignoring the discrete-continuous nature of the data on the estimation of extreme dependencies.

In this work, the Gaussian copula was included as a benchmark to evaluate its performance against more flexible alternatives, such as R-vine copulas, which better capture dependencies in zero-inflated data. Our results show that ignoring intermittency leads to an underestimation of extreme dependencies, aligning with findings in the literature and highlighting the limitations of this approach. While applying copulas to zero-inflated data without adjustments introduces biases, this model serves as a useful reference to demonstrate the importance of incorporating intermittency explicitly.

To clarify this aspect in the manuscript, we will expand the discussion on these limitations and explicitly compare this approach with models that account for zero inflation, ensuring that its role within our analysis is well understood.

**Model 3 (Vine Gaussian Copulas):**

**Are vine Gaussian copulas equivalent to traditional Gaussian copulas? When bivariate Gaussian copulas are assigned to the edges of a vine, the resulting multivariate density corresponds to a Gaussian density parameterized by a partial correlation vine rather than a standard correlation matrix. Clarification on this point would enhance understanding.**

Vine Gaussian copulas differ from traditional Gaussian copulas in their parametrization and dependency structure. The distinction lies in how dependence is structured: while Gaussian copulas rely on a single correlation matrix to describe dependencies, vine copulas decompose the joint distribution into a cascade of pair-copulas, allowing for more flexible and localized dependency modeling. This hierarchical structure enables vine Gaussian copulas to better capture complex dependence patterns while maintaining analytical tractability and computational efficiency in high-dimensional settings (Aas et al. 2009).

In our study, the vine Gaussian copula served as a baseline for evaluating how different copula structures represent multivariate dependencies. To ensure precision, we will revise the manuscript to explicitly differentiate vine Gaussian copulas from traditional Gaussian copulas.

**Model 4 (Vine Extreme Copula Model):**

**The terminology for this model may be misleading. It appears to be a vine copula where pair-copulas can be selected from different classes, including those that describe asymptotic tail dependence. This does not necessarily qualify it as an extreme-value copula. A reconsideration of the terminology is recommended to avoid confusion.**

The concern regarding the terminology is valid. The model employs an R-vine structure that allows for a flexible selection of bivariate copulas, including t-Student, Clayton, Gumbel, Joe6, BB1, BB6, BB7, BB8, and independent copulas, as well as rotated versions of Archimedean copulas to capture negative dependencies. While several of these copulas can model asymmetries and tail dependencies, we recognize that the term "Vine Extreme" could be misinterpreted as implying that the entire model is an extreme-value copula.

To avoid potential ambiguity, we will revise the terminology in the manuscript. We propose replacing *Vine Extreme Copula Model* with *Flexible Vine Copula Model*, which better reflects the model's composition without implying adherence to extreme-value theory.

**Figure 1:**

**It appears that model estimation is conducted independently within each group. Is this correct? Based on Equation (1), the model suggests conditional independence when conditioned on the rain/no-rain status for each site. As a result, the copula for Group 32 is entirely independent from the copula for Group 31. Is this assumption realistic? Further discussion on this matter is recommended.**

Model estimation is indeed conducted independently within each group, as illustrated in Figure 1 and defined in Equation (1). Each group contains all the events that generate precipitation in some stations, but not in others -except for Group 32, in which all events generate precipitation in all the stations-. Each group is represented as a binary sequence, where 1 indicates that all the events belonging to the group generate rainfall for that station, and 0 denotes the opposite behavior. The following sequences illustrate different distribution patterns:

- **Group 1** = [1,0,0,0,0] → Includes events where precipitation occurs only at the last station.

- **Group 2** = [1,1,0,0,0] → Includes events where precipitation occurs at the last two stations.

- **Group 3** = [1,1,1,0,0] → Includes events where precipitation occurs at the last three stations.

- …

- **Group 32** = [1,1,1,1,1] → Includes events where precipitation occurs at all stations.

However, while estimation is performed separately for each group, the joint cumulative distribution function (JCDF) integrates all groups collectively. This step is crucial for defining the critical hypersurface, as it captures the overall complexity of the hydrological system by combining probabilities across all possible precipitation configurations. The JCDF is not solely dependent on within-group dependencies but rather reflects the accumulated probability of extreme event occurrences across the full range of rainfall scenarios.

This strategy balances computational efficiency with the need to preserve the system's spatial and temporal complexity. We propose the following modification to the manuscript discussion, as this aspect deserves further attention:

*"While model estimation is conducted independently for each group, considering specific configurations of rainfall occurrence or absence across the stations, the construction of the joint cumulative distribution function (JCDF) is based on the integration of all modeled groups. This approach allows for capturing the complexity of the hydrological system by combining the probabilities associated with each configuration within a unified framework.*

*Segmenting by groups facilitates detailed modeling of particular scenarios, while the JCDF, calculated by summing the conditional probabilities of each group, preserves the system's integrity by considering all possible scenarios. This approach is particularly relevant in defining the critical surface, where it is necessary to evaluate the accumulated probability of extreme event occurrence across all possible combinations of rainfall and no rainfall among the stations.*

*Although this method assumes conditional independence between groups during estimation, the final integration into the joint CDF enables the evaluation of dependencies at a global level. This combination of segmented modeling and joint analysis aims to balance computational efficiency with an accurate representation of the system's dependency structure."*

**Page 11 (Marginal GEV Estimation):**

**The manuscript briefly mentions that marginal GEV estimation is conducted using Bayesian techniques. However, details regarding the Bayesian approach are sparse. Please provide a more comprehensive description of the estimation procedure.**

The estimation of Generalized Extreme Value (GEV) distribution parameters was conducted using Bayesian techniques, implemented via the Stan package in Python, which leverages Markov Chain Monte Carlo (MCMC) algorithms to model complex statistical structures. This approach accounts for parameter uncertainty, providing a more robust representation of variability in extreme events.

For each station, non-informative priors were assigned to the location, scale, and shape parameters, ensuring that posterior estimates were primarily data-driven. The No-U-Turn Sampler (NUTS) algorithm, integrated into Stan, was used for MCMC simulations, optimizing parameter space exploration and improving sampling convergence.

Recognizing that the main manuscript provides only a brief mention of this procedure, we will expand the description in the Supplementary Information, detailing:

- Bayesian Framework

- The rationale for the choice of prior distributions.

- Bayesian Parameter Estimation and MCMC Implementation for GEV Models

- Convergence validation criteria and goodness-of-fit metrics.

This addition will enhance methodological transparency and facilitate reproducibility for researchers applying Bayesian techniques in extreme value modeling.

**Line 310 (Computational Complexity of GPR Technique):**

**Given the computational intensity of calculating copula values from vine specifications, more details should be provided, including an algorithm if possible, to clarify the practical implementation of the Gaussian Process Regression (GPR) technique.**

The comment on the computational complexity of Gaussian Process Regression (GPR), particularly in relation to calculating copula values from vine specifications, is well taken. A more detailed explanation of its implementation and role in reducing computational costs will be included in the manuscript.

As previously noted, GPR is central to our methodology, providing an efficient surrogate model for approximating the joint distribution function while avoiding repeated, computationally intensive evaluations of vine copula structures. This approach effectively manages the high-dimensional complexity of copula-based models, particularly in calculating the critical layer associated with joint return periods.

To ensure transparency and reproducibility, a detailed description of the GPR methodology will be included in the Supplementary Information, covering:

- Theoretical foundations of GPR in the context of copula modeling.
- Detailed algorithmic steps, including data preparation, training, and validation.
- Hyperparameter selection and kernel optimization processes.
- Application to JCDF Estimation.

**Line 330 (Upper Tail Dependence):**

**The statement regarding the superior fit of copulas with upper tail dependence requires clarification. The model is five-dimensional, making it unclear how tail dependence is conceptualized and evaluated. Further elaboration is necessary.**

The reviewer raises an important point regarding the conceptualization and evaluation of tail dependence in a five-dimensional setting. In our study, tail dependence is assessed at the pairwise level by calculating the tail dependence coefficient for each station pair. These coefficients are then analyzed collectively to infer patterns in the multivariate space. This approach enables us to identify consistent upper tail dependence structures across multiple locations rather than relying on a single aggregated metric for the entire five-dimensional model.

The statement regarding the superior fit of copulas with upper tail dependence is supported by both visual and statistical evidence. While KDE density plots (Figure 9) provide an intuitive representation of tail concentration, statistical validation was conducted using the non-parametric estimator proposed by Schmidt and Stadtmuller (2006), known for its ability to detect tail dependencies in multivariate contexts, as detailed in lines 203–205. This method, also applied in Brunner, Furrer, and Favre (2019), formally quantifies tail dependence and confirms its presence in the dataset. However, we are aware of the limitations of this method, as acknowledged in the manuscript by citing Serinaldi, Bárdossy, and Kilsby (2015), who note that the estimator can be sensitive to data sparsity in extreme tails, potentially affecting the stability of the estimates. Therefore, the results were interpreted with caution, taking into account both the statistical limitations of the method and prior evidence supporting the existence of upper tail dependence in extreme precipitation events (Serinaldi 2013; Evin, Favre, and Hingray 2018).

To address this comment, we will revise the manuscript to explicitly define how tail dependence is evaluated within the five-dimensional framework and refine the statement regarding the fit of copulas with upper tail dependence to ensure alignment with this explanation.

**Section 3.5.2 (Likelihood vs. Probability):**

**It may be beneficial to distinguish between likelihood and probability, as is customary in the literature. This would ensure greater terminological precision and alignment with established conventions.**

**I hope these suggestions prove helpful in strengthening the manuscript.**

The distinction between "likelihood" and "probability" is well noted. Referring to the algorithm as "Maximum Likelihood Estimation (MLE)" may be misleading, as it does not maximize a likelihood function in the classical sense but rather applies gradient descent to identify the highest-density point within the critical layer.

To enhance clarity, we will specify that, while the algorithm is named *"MLE"* following the convention used in the Spotpy library, its application in this context does not adhere to the statistical definition of likelihood. Rather, it is employed to optimize the joint probability density function and identify the most representative events within the defined space.

Proposed modification in the manuscript (*Section 3.5.2*):

*"Although referred to as the Maximum Likelihood Estimation (MLE) algorithm, following the naming convention used in the Spotpy library, this method does not perform classical likelihood maximization. Instead, it functions as a gradient descent optimization technique aimed at maximizing the joint probability density function. This distinction is essential, as the term 'likelihood' here reflects the algorithm's label rather than the statistical definition. This approach was selected due to its computational efficiency in identifying the most probable event within the critical layer."*

**References**

Aas, Kjersti, Claudia Czado, Arnoldo Frigessi, and Henrik Bakken. 2009. "Pair-Copula Constructions of Multiple Dependence." *Insurance: Mathematics and Economics* 44 (2): 182–98. https://doi.org/10.1016/j.insmatheco.2007.02.001.

Brunner, Manuela I. 2023. "Floods and Droughts: A Multivariate Perspective." *Hydrology and Earth System Sciences* 27 (13): 2479–97. https://doi.org/10.5194/hess-27-2479-2023.

Brunner, Manuela I., Reinhard Furrer, and Anne-Catherine Favre. 2019. "Modeling the Spatial Dependence of Floods Using the Fisher Copula." *Hydrology and Earth System Sciences* 23 (1): 107–24. https://doi.org/10.5194/hess-23-107-2019.

Evin, Guillaume, Anne-Catherine Favre, and Benoit Hingray. 2018. "Stochastic Generation of Multi-Site Daily Precipitation Focusing on Extreme Events." *Hydrology and Earth System Sciences* 22 (1): 655–72. https://doi.org/10.5194/hess-22-655-2018.

Schmidt, Rafael, and Ulrich Stadtmuller. 2006. "Non-Parametric Estimation of Tail Dependence." *Scandinavian Journal of Statistics* 33 (2): 307–35. https://doi.org/10.1111/j.1467-9469.2005.00483.x.

Serinaldi, Francesco. 2009. "Copula-Based Mixed Models for Bivariate Rainfall Data: An Empirical Study in Regression Perspective." *Stochastic Environmental Research and Risk Assessment* 23 (5): 677–93. https://doi.org/10.1007/s00477-008-0249-z.

———. 2013. "An Uncertain Journey Around the Tails of Multivariate Hydrological Distributions: Multivariate Frequency Analysis with Uncertainty." *Water Resources Research* 49 (10): 6527–47. https://doi.org/10.1002/wrcr.20531.

Serinaldi, Francesco, András Bárdossy, and Chris G. Kilsby. 2015. "Upper Tail Dependence in Rainfall Extremes: Would We Know It If We Saw It?" *Stochastic Environmental Research and Risk Assessment* 29 (4): 1211–33. https://doi.org/10.1007/s00477-014-0946-8.

**Return period of high-dimensional compound events. Part II: Analysis of spatially-variable precipitation - Supplemantary information**

Diego Urrea Méndez[1], Manuel Del Jesus[1], and Dina Vanessa Gomez Rave[1]

[1]IHCantabria - Instituto de Hidráulica Ambiental de la Universidad de Cantabria, Santander, Spain.

**Correspondence:** Manuel Del Jesus (manuel.deljesus@unican.es)

**1 Generalized extreme value distribution**

The Generalized Extreme Value (GEV) distribution, rooted in the work of Fisher and Tippett (1928), was formalized by Jenkinson (1955), who unified the Gumbel, Fréchet, and Weibull distributions into a single framework. Widely applied in hydrology, climatology, and engineering, the GEV is defined by three parameters: location ($\mu$), scale ($\sigma$), and shape ($\xi$), which control its position, spread, and tail behavior. Its flexibility in modeling light-tailed, heavy-tailed, and bounded distributions makes it a key tool for analyzing extreme events.

The probability density function (pdf) of the GEV is defined in Eq. (S1) as:

$$f(x|\mu,\sigma,\xi) = \frac{1}{\sigma}\left[1 + \xi\left(\frac{x-\mu}{\sigma}\right)\right]^{-\left(\frac{1}{\xi}+1\right)} \exp\left\{-\left[1 + \xi\left(\frac{x-\mu}{\sigma}\right)\right]^{-\frac{1}{\xi}}\right\} \tag{S1}$$

where $1 + \xi((x-\mu)/\sigma) > 0$.

The GEV encompasses the following classical extreme value distributions. Gumbel ($\xi = 0$) for exponentially tailed events, Fréchet ($\xi > 0$) for heavy-tailed events nad Weibull ($\xi < 0$) for bounded-tailed events.

**2 Bayesian Framework**

The modeling of extreme events has become increasingly relevant in various scientific domains, including hydrology, climatology, and finance. Understanding and predicting rare but impactful events such as floods, heatwaves, and financial crashes is essential for effective risk management. The Generalized Extreme Value (GEV) distribution has emerged as a fundamental tool in this context due to its flexibility in modeling block maxima and its solid theoretical foundation derived from the Fisher-Tippett-Gnedenko theorem.

The Bayesian framework offers a robust approach to estimating GEV parameters, allowing the incorporation of prior knowledge and a natural quantification of uncertainty. This approach has been widely applied across different fields. In urban

planning, Balbi et al. (2016) applied a spatial Bayesian network model to assess the benefits of early warning systems for urban flood risk, improving risk management strategies in complex hydrological environments.

These diverse applications highlight the versatility and robustness of the Bayesian GEV approach in both environmental and economic contexts, underscoring its value as a key tool for decision-making under uncertainty.

The Bayesian approach allows incorporating prior information and quantifying uncertainty in parameter estimation. It is based on Bayes' theorem, presented in Eq. (S2):

$$p(\theta|x) = \frac{L(x|\theta)p(\theta)}{p(x)} \tag{S2}$$

where $p(\theta|x)$ is the posterior distribution, $L(x|\theta)$ is the likelihood function derived from the GEV, $p(\theta)$ represents the prior distributions assigned to the parameters, and $\theta = (\mu, \sigma, \xi)$ represents the vector of unknown parameters. The term $p(x)$, known as the evidence or marginal likelihood, acts as a normalizing constant ensuring that the posterior distribution integrates to one. Although $p(x)$ is independent of $\theta$ and often omitted in practical MCMC implementations, it is essential in the formal definition of Bayes' theorem.

Given the analytical intractability of the posterior distribution, Markov Chain Monte Carlo (MCMC) methods are used for posterior sampling. Algorithms such as Metropolis-Hastings and Hamiltonian Monte Carlo (HMC) are preferred due to their efficiency in exploring parameter spaces. These techniques allow for the estimation of the full posterior distribution of the GEV parameters, enabling a comprehensive uncertainty assessment that is critical for risk analysis in fields such as hydrology and finance.

Significant contributions to Bayesian GEV modeling have shaped the current state of the art. Coles and Powell (1996) laid the foundational work, introducing Bayesian principles into extreme value modeling and demonstrating their advantages over traditional methods. This was further extended by Heffernan and Tawn (2004), who incorporated covariates and hierarchical structures into the Bayesian GEV framework, enabling more complex and context-sensitive analyses.

**S2.1 Bayesian Parameter Estimation and MCMC Implementation for GEV Models**

The Bayesian fitting of the Generalized Extreme Value (GEV) distribution relies on carefully chosen prior distributions for its three parameters. In our analysis of rainfall extremes, the priors have been explicitly defined based on prior knowledge and the expected behavior of extreme precipitation events.

**S2.1.1 Prior Specifications**

Location Parameter ($\mu$): We assign a normal prior to $\mu$, formally defined in Eq. (S3):

$$\mu \sim \mathcal{N}(\mu_0, \sigma_0^2) \tag{S3}$$

In our analysis, we set $\mu_0$ to the sample mean of the annual maximum rainfall observations, reflecting our empirical understanding of the central tendency of extreme rainfall events. We choose $\sigma_0^2$ as a large value (e.g., 10 times the sample variance) to ensure that the prior is weakly informative, allowing the data to play a significant role in shaping the posterior estimates.

Scale Parameter ($\sigma$): We assign a log-normal prior to $\sigma$ by applying a log-transformation, formally defined in Eq. (S4):

$$\log(\sigma) \sim \mathcal{N}(\alpha, \beta^2) \tag{S4}$$

In our analysis, we set $\alpha = 0$ to center the distribution and choose $\beta^2$ as a large value (e.g., approximately 10, which corresponds to $\beta \approx 3$) to ensure weak informativeness. This approach guarantees that $\sigma$ remains positive and allows for a wide range of plausible values, enabling the data to have a significant influence on the posterior estimates. Additionally, the log-transformation improves computational stability and chain mixing during the MCMC sampling.

Shape Parameter ($\xi$):We assign a normal prior to $\xi$, formally defined in Eq. (S5):

$$\xi \sim \mathcal{N}(0, \gamma^2) \tag{S5}$$

In our analysis, we set $\gamma = 0.25$, which concentrates most of the prior mass in the interval $[-0.5, 0.5]$. Given the high sensitivity of the shape parameter to prior specifications, this choice reflects our empirical expectations regarding tail behavior in extreme rainfall events and allows the model to adequately capture the three types of tail behavior (Gumbel, Fréchet, and Weibull). This careful selection ensures that the data play a significant role in refining the posterior estimates for $\xi$.

The shape parameter exhibits the highest sensitivity to the prior specification, directly influencing the inference on extreme values. In contrast, the location parameter tends to be robust across different prior choices, while the scale parameter shows moderate sensitivity. For the estimation, MCMC methods are employed; the log-transformation of $\sigma$ improves both computational stability and chain mixing.

This explicit framework for prior selection balances theoretical requirements with practical implementation, thereby providing a reliable foundation for extreme value analysis in the context of rainfall extremes.

The likelihood function is formulated based on the GEV density and the observed data, emphasizing the accurate representation of tail behavior, which is crucial in extreme value analysis.

To estimate the posterior distribution, MCMC methods are employed, offering a robust alternative to conventional numerical techniques. In this study, the PyStan library was used for Bayesian inference, utilizing four independent chains of length $N = 1000$, following a burn-in period of 2000 iterations to ensure convergence. The Metropolis-Hastings algorithm was applied to construct the sampling chains, enabling efficient exploration of the parameter space and reducing autocorrelation between samples.

Convergence diagnostics were assessed using the R-hat statistic, where values close to 1 indicate satisfactory convergence across chains. Finally, the posterior analysis focused on estimating key statistics such as the mean, median, and mode for each

parameter, alongside constructing 95% credible intervals to quantify uncertainty. These intervals represent the range within
which the true parameter values are expected to lie with 95% probability, offering a comprehensive understanding of the
parameter space under the Bayesian framework.

**3   Gaussian Process Regression (GPR)**

Gaussian Process Regression (GPR) is a flexible, non-parametric probabilistic modeling approach widely used for regression
tasks involving complex, non-linear relationships. It was formalized for machine learning applications by Rasmussen and
Williams (2008), GPR models data as a distribution over functions, defined by a mean function and a covariance function
(kernel). Typically, the mean function is set to zero, focusing attention on the choice of the covariance function, which encodes
the assumptions about the function's smoothness, periodicity, and overall behavior.

The core of GPR lies in the specification of the kernel function, which determines the covariance between data points. In our
study, the *Rational Quadratic kernel* was identified as the most suitable for capturing the underlying relationships within the
data. This kernel combines the properties of the Radial Basis Function (RBF) and a scale mixture of Gaussian kernels, allowing
it to handle varying length scales effectively. This flexibility was crucial when modeling the complex dependence structures of
compound precipitation events across multiple locations.

The Gaussian Process is formally expressed as:

$$f(x) \sim \mathcal{GP}(m(x), k(x, x')) \tag{S6}$$

where $m(x)$ is the mean function (often zero) and $k(x, x')$ is the covariance function, or kernel. Several kernel functions can
be used in GPR, each offering different properties in terms of smoothness and flexibility:

1. **Radial Basis Function (RBF) Kernel**: Also known as the Squared Exponential kernel, it assumes smooth and infinitely
   differentiable functions. It is defined as:

$$k_{\text{RBF}}(x, x') = \sigma^2 \exp\left(-\frac{(x - x')^2}{2l^2}\right) \tag{S7}$$

where $\sigma^2$ is the variance and $l$ is the length scale controlling the smoothness.

2. **Matérn Kernel**: Provides more flexibility by controlling the smoothness through the parameter $\nu$. For $\nu = \frac{3}{2}$ or $\nu = \frac{5}{2}$,
   the kernel has specific closed forms. The general form is:

$$k_{\text{Matérn}}(x, x') = \sigma^2 \frac{2^{1-\nu}}{\Gamma(\nu)} \left(\frac{\sqrt{2\nu}|x - x'|}{l}\right)^\nu K_\nu\left(\frac{\sqrt{2\nu}|x - x'|}{l}\right) \tag{S8}$$

where $K_\nu$ is the modified Bessel function of the second kind, $l$ is the length scale, and $\nu$ controls smoothness.

3. **Rational Quadratic Kernel**: Acts as a scale mixture of RBF kernels with different length scales, offering robustness in modeling varying degrees of smoothness. It is defined as:

$$k_{\mathrm{RQ}}(x, x') = \sigma^2 \left(1 + \frac{(x - x')^2}{2\alpha l^2}\right)^{-\alpha} \tag{S9}$$

where $\alpha$ is a scale-mixture parameter controlling the relative weighting of small and large-scale variations.

**S3.0.1 Application to Copula CDF Estimation**

In our framework, GPR was employed to approximate the Joint Cumulative Distribution Function (JCDF), a critical step in understanding the joint behavior of multiple variables or, in our case, a single variable measured at different locations. Direct computation of the JCDF, especially for high-dimensional data and complex copula structures like R-vines, is computationally expensive. To address this, GPR served as a surrogate model, offering a balance between computational efficiency and accuracy.

The process begins by generating a representative sample of data points from the JCDF using conventional methods, ensuring sufficient coverage of the distribution's variability. These samples serve as the training dataset for the GPR model. The Rational Quadratic kernel is often preferred for its flexibility in adapting to varying data scales, effectively capturing both local and global dependencies within the JCDF structure.

Once trained, the GPR model was used to predict JCDF values for a much larger dataset, enabling high-resolution analysis without incurring significant computational costs. The predictive mean from the GPR provided the JCDF approximation, while the predictive variance quantified the uncertainty of the estimation.

The hyperparameters of the GPR, including the length scale ($l$), variance ($\sigma^2$), and the mixture parameter ($\alpha$) for the Rational Quadratic kernel, were optimized by maximizing the log-marginal likelihood:

$$\log p(y|X) = -\frac{1}{2} y^T [K(X, X) + \sigma_n^2 I]^{-1} y - \frac{1}{2} \log |K(X, X) + \sigma_n^2 I| - \frac{n}{2} \log 2\pi \tag{S10}$$

This optimization ensured that the GPR achieved a balance between model complexity and data fit, preventing overfitting while capturing essential dependencies.

The predictive performance was evaluated using both relative and absolute errors, with the Rational Quadratic kernel achieving the lowest error rates among the tested kernels. Error values consistently remained below 0.05, validating the effectiveness of the GPR in approximating the JCDF.

By leveraging GPR, we significantly reduced the computational burden of high-dimensional copula analysis, enabling efficient calculation of Joint Return Periods (JRP) and critical layers. This approach provided a robust and scalable framework for extreme value analysis in hydrology.

.

**References**

Balbi, S., Villa, F., Mojtahed, V., Hegetschweiler, K. T., and Giupponi, C.: A spatial Bayesian network model to assess the benefits of early warning for urban flood risk to people, Natural Hazards and Earth System Sciences, 16, 1323–1337, https://doi.org/10.5194/nhess-16-1323-2016, publisher: Copernicus GmbH, 2016.

Coles, S. G. and Powell, E. A.: Bayesian Methods in Extreme Value Modelling: A Review and New Developments, International Statistical Review / Revue Internationale de Statistique, 64, 119–136, https://doi.org/10.2307/1403426, publisher: [Wiley, International Statistical Institute (ISI)], 1996.

Fisher, R. A. and Tippett, L. H. C.: Limiting forms of the frequency distribution of the largest or smallest member of a sample, Mathematical Proceedings of the Cambridge Philosophical Society, 24, 180–190, https://doi.org/10/bscxvn, publisher: Cambridge University Press, 1928.

Heffernan, J. E. and Tawn, J. A.: A conditional approach for multivariate extreme values (with discussion), Journal of the Royal Statistical Society: Series B (Statistical Methodology), 66, 497–546, https://doi.org/10.1111/j.1467-9868.2004.02050.x, _eprint: https://onlinelibrary.wiley.com/doi/pdf/10.1111/j.1467-9868.2004.02050.x, 2004.

Jenkinson, A. F.: The frequency distribution of the annual maximum (or minimum) values of meteorological elements, Quarterly Journal of the Royal Meteorological Society, 81, 158–171, https://doi.org/10.1002/qj.49708134804, _eprint: https://onlinelibrary.wiley.com/doi/pdf/10.1002/qj.49708134804, 1955.

Rasmussen, C. E. and Williams, C. K. I.: Gaussian processes for machine learning, Adaptive computation and machine learning, MIT Press, Cambridge, Mass., 3. print edn., ISBN 978-0-262-18253-9, 2008.